# From gigawatt to multi-gigawatt wind farms: wake effects, energy budgets and inertial gravity waves investigated by large-eddy simulations

**Oliver Maas**

Institute of Meteorology and Climatology, Leibniz University Hannover, Hanover, Germany

**Correspondence:** Oliver Maas (maas@meteo.uni-hannover.de)

**Abstract.** The size of newly installed offshore wind farms increases rapidly. Planned offshore wind farm clusters have a rated capacity of several gigawatts and a length of up to 100 km. The flow through and around wind farms of this scale can be significantly different than the flow through and around smaller wind farms on the sub-gigawatt scale. A good understanding of the involved flow physics is vital for accurately predicting the wind farm power output as well as predicting the meteorological conditions in the wind farm wake. To date there is no study that directly compares small wind farms (sub-gigawatt) with large wind farms (super-gigawatt) in terms of flow effects or power output. The aim of this study is to fill this gap by providing this direct comparison by performing large-eddy simulations of a small wind farm (13 km length) and a large wind farm (90 km length) in a convective boundary layer, which is the most common boundary layer type in the North Sea.

The results show that there are significant differences in the flow field and the energy budgets of the small and large wind farm. The large wind farm triggers an inertial wave with a wind direction amplitude of approximately $10°$ and a wind speed amplitude of more than $1\,\mathrm{m\,s^{-1}}$. In a certain region in the far wake of a large wind farm the wind speed is greater than far upstream of the wind farm, which can be beneficial for a downstream located wind farm. The inertial wave also exists for the small wind farm, but the amplitudes are approximately 4 times weaker and thus may be hardly observable in real wind farm flows that are more heterogeneous. Regarding turbulence intensity, the wake of the large wind farm has the same length as the wake of the small wind farm and is only a few kilometers long. Both wind farms trigger inertial gravity waves in the free atmosphere, whereas the amplitude is approximately twice as large for the large wind farm. The inertial gravity waves induce streamwise pressure gradients inside the boundary layer, affecting the energy budgets of the wind farms. The most dominant energy source of the small wind farm is the horizontal advection of kinetic energy, but for the large wind farm the vertical turbulent flux of kinetic energy is 5 times greater than the horizontal advection of kinetic energy. The energy input by the gravity-wave-induced pressure gradient is greater for the small wind farm because the pressure gradient is greater. For the large wind farm, the energy input by the geostrophic forcing (synoptic-scale pressure gradient) is significantly enhanced by the wind direction change that is related to the inertial oscillation. For both wind farms approximately 75 % of the total available energy is extracted by the wind turbines and 25 % is dissipated.

## 1 Introduction

The size of newly installed offshore wind farms increases rapidly. The largest wind farm in operation Moray East (United Kingdom) has a rated capacity of 950 MW and consists of 100 wind turbines (Herzig, 2022). The largest wind farm under construction is Hollandse Kust Zuid (Netherlands), with a rated capacity of 1540 MW. It consists of 140 wind turbines and has a length of approximately 15 km (Herzig, 2022). Offshore wind farms are often arranged in clusters, so that the cluster capacity can already be in the multi-gigawatt scale. One example is the planned wind farm cluster in Zone 3 in the German Bight with a planned capacity of 20 GW and a length of approximately 100 km (BSH, 2021).

The flow through and around wind farms of this scale can be significantly different than the flow through and around smaller wind farms on the sub-gigawatt scale, as recently published results show. For example, large wind farms can cause a significant counterclockwise wind direction change in the wake and a vertical displacement of the inversion layer above the wind farm (Allaerts and Meyers, 2016; Lanzilao and Meyers, 2022; Maas and Raasch, 2022). A good understanding of the involved flow physics is vital for accurately predicting the wind farm power output as well as predicting the meteorological conditions in the wind farm wake. The "improved understanding of atmospheric and wind power plant flow physics" is stated as one of the grand challenges in the science of wind energy by Veers et al. (2019) because the involved scales range from microscale to mesoscale and interactions can be complex. The best numerical method for the investigation of these interactions that considers all relevant physical processes but is still computationally feasible is large-eddy simulation (LES).

In recent years many LES studies investigated wind farm flows. The studies can be subdivided into three categories. The first category investigates infinitely large wind farms by using cyclic boundary conditions in both lateral directions, e.g., Abkar and Porté-Agel (2013), Abkar and Porté-Agel (2014), Calaf et al. (2010), Calaf et al. (2011), Johnstone and Coleman (2012), Lu and Porté-Agel (2011), Lu and Porté-Agel (2015), Meyers and Meneveau (2013), Porté-Agel et al. (2014) and VerHulst and Meneveau (2014). The second category investigates semi-infinite wind farms by using cyclic boundary conditions only in the crosswise direction, e.g., Allaerts and Meyers (2016), Allaerts and Meyers (2017), Allaerts and Meyers (2018), Andersen et al. (2015), Centurelli et al. (2021), Segalini and Chericoni (2021), Stevens et al. (2016), Wu and Porté-Agel (2017) and Zhang et al. (2019). The third category investigates wind farms that have a finite size in both lateral directions which also include real wind farms, e.g., Dörenkämper et al. (2015), Ghaisas et al. (2017), Lanzilao and Meyers (2022), Maas and Raasch (2022), Nilsson et al. (2015), Porté-Agel et al. (2013), Witha et al. (2014) and Wu and Porté-Agel (2015).

Typical wind farm lengths in the semi-infinite wind farm studies range from 5 km (Centurelli et al., 2021) over 15 km (Allaerts and Meyers, 2017) to 24 km (Andersen et al., 2015). Typical wind farm lengths in the finite-size wind farm studies range from 2 km (Witha et al., 2014) over 15 km (Lanzilao and Meyers, 2022) to approximately 100 km (Maas and Raasch, 2022). Thus, most of the studies are representative for existing, state-of-the-art wind farms and do not represent the spatial scales that future wind farm clusters will have. Specifically, there is no study that directly compares small wind farms (10 km scale) with large wind farms (100 km scale) in terms of flow effects or power output, neither with LESs nor with simpler models.

The aim of this study is to provide this direct, systematic comparison by performing LESs of a small wind farm (13 km length) and a large wind farm (90 km length) with a semi-infinite wind farm setup. The comparison focuses on the boundary layer flow inside the wind farm but also in the far wake and the overlying free atmosphere. A detailed energy budget analysis is made to identify the dominant energy sources and sinks for small and large wind farms. The domain is more than 400 km long to cover the far wake and has a height of 14 km to cover wind-farm-induced gravity waves. The boundary layer is filled with a turbine-wake-resolving grid resulting in more than 2 billion grid points in total.

The paper is structured as follows. The numerical model and the main and precursor simulations are described in Sect. 2. The simulation results are presented in Sect. 3, and Sect. 4 concludes and discusses the results of the study.

## 2 Methods

### 2.1 Numerical model

The simulations are carried out with the Parallelized Large-eddy Simulation Model (PALM; Maronga et al., 2020). PALM is developed at the Institute of Meteorology and Climatology of Leibniz Universität Hannover, Germany. Several wind farm flow investigations have been successfully conducted with this code in the past (e.g., Witha et al., 2014; Dörenkämper et al., 2015; Maas and Raasch, 2022). PALM solves the non-hydrostatic, incompressible Navier–Stokes equations in Boussinesq-approximated form. The equations for the conservation of mass, momentum and internal energy then read as

$$\frac{\partial \tilde{u}_j}{\partial x_j} = 0, \tag{1}$$

$$\frac{\partial \tilde{u}_i}{\partial t} = -\frac{\partial \tilde{u}_i \tilde{u}_j}{\partial x_j} - \epsilon_{ijk} f_j \tilde{u}_k + \epsilon_{i3j} f_3 u_{\mathrm{g},j} - \frac{1}{\rho_0} \frac{\partial \pi^*}{\partial x_i}$$
$$+ g \frac{\tilde{\theta} - \langle \tilde{\theta} \rangle}{\langle \tilde{\theta} \rangle} \delta_{i3} - \frac{\partial}{\partial x_j} \left( \widetilde{u_i'' u_j''} - \frac{2}{3} e \delta_{ij} \right) + d_i, \tag{2}$$

$$\frac{\partial \tilde{\theta}}{\partial t} = -\frac{\partial \tilde{u}_j \tilde{\theta}}{\partial x_j} - \frac{\partial}{\partial x_j} \left( \widetilde{u_j'' \theta''} \right), \tag{3}$$

where angular brackets indicate horizontal averaging and a double prime indicates subgrid-scale (SGS) quantities; a tilde denotes filtering over a grid volume; $i, j, k \in \{1, 2, 3\}$, $u_i$, $u_j$, $u_k$ are the velocity components in the respective directions ($x_i, x_j, x_k$); $\theta$ is potential temperature; $t$ is time; and $f_i = (0, 2\Omega\cos(\phi), 2\Omega\sin(\phi))$ is the Coriolis parameter with the Earth's angular velocity $\Omega = 0.729 \times 10^{-4}\,\mathrm{rad\,s^{-1}}$ and the geographical latitude $\phi$. The geostrophic wind speed components are $u_{g,j}$, and the basic state density of dry air is $\rho_0$. The modified perturbation pressure is $\pi^* = p + \frac{2}{3}\rho_0 e$, where $p$ is the perturbation pressure and $e = \frac{1}{2}\widetilde{u_i'' u_i''}$ is the SGS turbulence kinetic energy. The gravitational acceleration is $g = 9.81\,\mathrm{m\,s^{-2}}$, $\delta$ is the Kronecker delta and $d_i$ represents the forces of the wind turbine actuator discs.

The SGS model uses a 1.5-order closure according to Deardorff (1980), modified by Moeng and Wyngaard (1988) and Saiki et al. (2000). The wind turbines are represented by an advanced actuator disc model with rotation (ADM-R) that acts as an axial momentum sink and an angular momentum source (inducing wake rotation). The ADM-R is described in detail by Wu and Porté-Agel (2011) and was implemented in PALM by Steinfeld et al. (2015). Additional information is also given by Maas and Raasch (2022). The wind turbines have a yaw controller that aligns the rotor axis with the wind direction.

## 2.2 Main simulations

The study consists of two simulations. The first simulation contains a small wind farm with $N_x \times N_y = 8 \times 8 = 64$ wind turbines resulting in a length of 13.44 km. The second simulation contains a large wind farm with $N_x \times N_y = 48 \times 8 = 384$ wind turbines resulting in a length of 90.24 km (see Fig. 1). The wind farms extend over the entire domain width, and cyclic boundary conditions are applied in the $y$ direction, so that the wind farms are effectively infinitely large in this direction. This idealized setup has been used in many other LES wind farm studies, e.g., Stevens et al. (2016), Allaerts and Meyers (2017) and Wu and Porté-Agel (2017). It simplifies the data analysis and allows us to focus only on streamwise variations in the wind farm and the wake. The validity of the results for finite-size, real wind farms is discussed in Sect. 4.

The IEA 15 MW wind turbine with a rotor diameter of $D = 240$ m and a rated power of 15 MW is used (Gaertner et al., 2020). The hub height is set to 180 m instead of 150 m, so that the turbulent fluxes at the rotor bottom are better resolved by the numerical grid. The wind turbines are arranged in a staggered configuration and have a streamwise and crosswise spacing of $s = 8\,D$, resulting in an installed capacity density of $4.07\,\mathrm{W\,m^{-2}}$. The small wind farm has a length of 13.44 km, which corresponds approximately to the length of the currently largest wind farm under construction, Hollandse Kust Zuid. The large wind farm has a length

of 90.24 km, which corresponds approximately to the length of the planned wind farm cluster in Zone 3 in the German Bight. Note that the small wind farm is already as long as the largest wind farms of most other LES studies, e.g., Wu and Porté-Agel (2017) (19.6 km) and Allaerts and Meyers (2017) (15 km).

The domain has a length of $L_x = 409.6$ km to cover the far wake of the wind farms. The wind farms have a distance of 100 km to the inflow boundary, so that the wind-farm-induced flow blockage is covered. The domain width is $L_y = 15.36$ km for the small and large wind farm case. A domain height of $L_z = 14.0$ km is required to cover wind-farm-induced gravity waves. That the Boussinesq approximation is still valid for such a large domain height is shown in Appendix A. To avoid reflection of the waves at the domain top, there is a Rayleigh damping layer above $z_{\mathrm{rd}} = 5$ km. The Rayleigh damping factor increases from zero at the bottom of the damping layer to its maximum value of $f_{\mathrm{rdm}} = 0.025(\Delta t)^{-1} \approx 0.017\,\mathrm{s^{-1}}$ at the domain top according to this function (see Fig. 1):

$$f_{\mathrm{rd}}(z) = f_{\mathrm{rdm}}\sin^2\left(0.5\pi\frac{z - z_{\mathrm{rd}}}{L_z - z_{\mathrm{rd}}}\right). \tag{4}$$

This sine wave profile leads to fewer reflections compared to a linear profile (Klemp and Lilly, 1978). The choice of these parameters is based on a set of test simulations with a larger grid spacing that are performed to find parameters that result in a low reflectivity. The reflectivity is obtained by the method described by Allaerts and Meyers (2017), which is a modified version of the method described by Taylor and Sarkar (2007). With the chosen parameters, less than 6 % of the upwards propagating wave energy is reflected.

The domain is filled with an equidistant regular grid with a grid spacing of 20 m, yielding a density of 12 grid points per rotor diameter. This is enough to resolve the most relevant eddies inside the wind turbine wakes. Steinfeld et al. (2015) showed that even eight grid points per rotor diameter are sufficient to obtain a converged result for the mean wind speed profiles at a downstream distance of $5D$. Above 900 m the grid is vertically stretched by 8 % per grid point up to a maximum vertical grid spacing of 200 m, which is enough for resolving the gravity waves with a vertical wavelength of approximately 5 km (see Table 1 in Sect. 3.4). For the small wind farm the grid contains $n_x \times n_y \times n_z = 20480 \times 768 \times 128 \approx 2.01 \times 10^9$ grid points, and for the large wind farm the grid contains $n_x \times n_y \times n_z = 20480 \times 1536 \times 128 \approx 4.01 \times 10^9$ grid points CE1.

The flow field is initialized by the instantaneous flow field of the last time step of a precursor simulation. Details about the precursor simulation and the meteorological parameters are given in the next section. The flow field is filled cyclically into the main domain because it is larger than the precursor domain. At the inflow, vertical velocity and temperature profiles averaged over the last 2 h of the precursor simulation are prescribed. The turbulent state of the in-

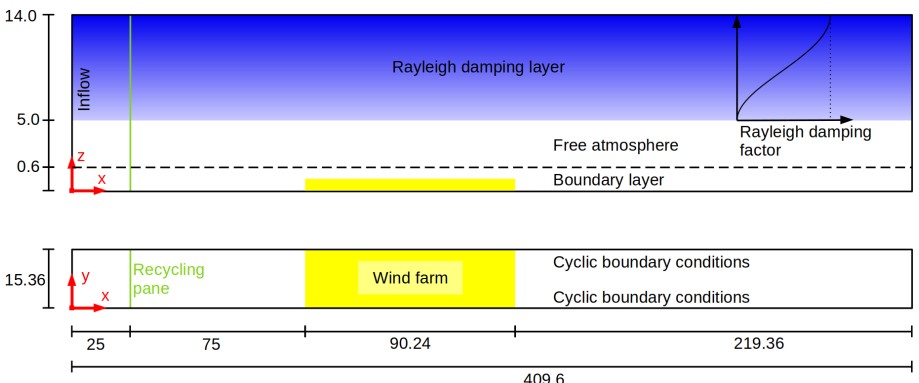

**Figure 1.** Side view and plan view of the domain and wind farm layout for the large wind farm case. Dimensions in kilometers.

flow is maintained by a turbulence recycling method that maps the turbulent fluctuations from the recycling plane at $x = 25$ km onto the inflow plane at $x = 0$. Details of the recycling method are given in Maas and Raasch (2022). The large distance between inflow and recycling plane is chosen to cover elongated convection rolls that appear in the convective boundary layer (CBL) and to cover at least twice the advection distance of the convective timescale $U_g z_i / w_* = 9.011 \, \mathrm{m\,s^{-1}} \, 600 \, \mathrm{m} / 0.49 \, \mathrm{m\,s^{-1}} \approx 11$ km, with the convective velocity scale $w_* = \left[ \frac{g z_i}{\overline{\theta}} \langle \overline{w'\theta'} \rangle_s \right]^{1/3}$, where $\overline{\theta} = 280$ K and $\langle \overline{w'\theta'} \rangle_s$ is the horizontally averaged kinematic surface heat flux averaged over the last 4 h of the precursor simulation. For the potential temperature, the absolute value is recycled instead of the turbulent fluctuation, so that the inflow temperature increases according to the surface temperature. The otherwise constant inflow temperature profile would cause a streamwise temperature gradient that triggers a thermal circulation inside the entire domain. The turbulent fluctuations are shifted in the $y$ direction by $+6.4$ km to avoid streamwise streaks in the averaged velocity fields; for further details please refer to Maas and Raasch (2022) and Munters et al. (2016). Radiation boundary conditions as described by Miller and Thorpe (1981) and Orlanski (1976) are used at the outflow plane. Hereby, the flow quantity $q$ at the outflow boundary $b$ is determined with the phase velocity $\hat{c}$ and the upstream derivative of the flow quantity:

$$q_b^{t+\Delta t} = q_b^t - (\hat{c} \Delta t / \Delta x)(q_b^n - q_{b-1}^n). \quad (5)$$

The phase velocity $\hat{c}$ is set to the maximum possible phase velocity of $\Delta x / \Delta t$. The surface boundary conditions and other parameters are the same as in the precursor simulation and are thus described in the next section. The physical simulation time of the main simulations is 20 h, and the presented data are averaged over the last 4 h.

## 2.3 Precursor simulation

Initial and inflow profiles of both simulations are obtained by a precursor simulation without a wind farm. It has cyclic boundary conditions in both lateral directions and a domain size of $L_{x,\mathrm{pre}} \times L_{y,\mathrm{pre}} \times L_{z,\mathrm{pre}} = 15.36 \times 9.6 \times 14.0 \, \mathrm{km^3}$. The number of vertical grid points, the vertical grid stretching and Rayleigh damping levels are the same as in the main simulation. The initial horizontal velocity is set to the geostrophic wind $(U_g, V_g) = (9.011, -1.527) \, \mathrm{m\,s^{-1}}$, resulting in a steady-state hub height wind speed of $9.0 \pm 0.02 \, \mathrm{m\,s^{-1}}$ that is aligned with the $x$ axis ($\pm 0.01°$). The values for the geostrophic wind are obtained by iterative adjustments between preliminary precursor simulations, of which two are needed to obtain the given accuracy. The latitude is $\phi = 55°$ N. The initial potential temperature is set to 280 K up to a height of 600 m and has a lapse rate of $\Gamma = +3.5 \, \mathrm{K\,km^{-1}}$ above. This lapse rate corresponds to the international standard atmosphere. The onset of turbulence is triggered by small random perturbations in the horizontal velocity field below a height of 300 m. A Dirichlet boundary condition is set for the surface temperature. Why a Dirichlet boundary condition is a good choice is explained in Maas and Raasch (2022). A constant surface heating rate of $\dot{\theta}_0 = +0.05 \, \mathrm{K\,h^{-1}}$ is applied, resulting in a Monin–Obukhov length of $L \approx -400$ m, which is common value for convective boundary layers in the North Sea (Muñoz-Esparza et al., 2012). The resulting boundary layer height (height of maximum vertical potential temperature gradient) of $z_i = 600$ m is a small but still typical value for convective boundary layers over the North Sea (Maas and Raasch, 2022). Boundary layer growth is avoided by applying a large-scale subsidence that acts only on the potential temperature field. The subsidence velocity is zero at the surface and increases linearly to its maximum value at $z = 600$ m and is constant above. The maximum subsidence velocity is chosen in such a way that the temperature increase in the free atmosphere (FA) exactly matches the surface heating rate: $w_{\mathrm{sub}} = \dot{\theta}_0 / \Gamma \approx 14.3 \, \mathrm{m\,h^{-1}}$. The roughness length for momentum and heat is $z_0 = z_{0,\mathrm{h}} = 1$ mm, and a constant flux layer is assumed between the surface and the lowest atmospheric grid level. At the domain top and bottom, a Neumann boundary condition for the perturbation pressure and Dirichlet boundary conditions for the

velocity components are used. For the potential temperature, a constant lapse rate is assumed at the domain top.

The physical simulation time of the precursor simulation is 48 h, to obtain a steady-state mean flow; i.e., the hourly-averaged hub height wind speed changes less than $0.05\,\mathrm{m\,s^{-1}}$ within 8 h. This long simulation time is needed for the decay of an inertial oscillation in time that has a period of 14.6 h. The inertial oscillation occurs because there is no equilibrium of forces in the boundary layer (BL) at the beginning of the simulation.

## 3 Results

### 3.1 Mean flow at hub height

To make a first qualitative comparison between the small and the large wind farm case, the mean horizontal wind speed and streamlines of the mean flow at hub height are shown in Fig. 2 for both cases. The most striking difference is the large modification of the wind direction that occurs for the large wind farm case. Inside the large wind farm the flow is deflected counterclockwise, but in the wake the flow is deflected clockwise so that the wind direction reaches the inflow wind direction and even turns further clockwise. But also for the wind speed both cases show significant differences. The wind speed reduction inside the large wind farm is significantly greater than inside the small wind farm, which is an expected result. Remarkable, however, is the fact that the wind speed in the far wake of the large wind farm is significantly greater than the inflow wind speed.

To make a more quantitative comparison between the two cases, Fig. 3 shows the mean horizontal wind speed, wind direction and perturbation pressure at hub height along $x$ for the small and large wind farm. The quantities are averaged along $y$ and a moving average with a window size of one turbine spacing is applied along $x$ to smooth out turbine-related sharp gradients. It can be seen that upstream of the wind farms the wind speed is reduced due to the blockage effect. The speed reduction $2.5\,D$ upstream of the first turbine row is 4.8 % for the small wind farm and 7.9 % for the large wind farm. These values lie in the range of 1 %–11 %, reported by Wu and Porté-Agel (2017) for a 20 km long wind farm under different FA stratifications. The blockage effect is caused by an increase in the perturbation pressure of 4.8 and 8.5 Pa relative to the pressure at the inflow for the small and large wind farm, respectively (see Fig. 3c). The perturbation pressure distribution is related to gravity waves that form in the free atmosphere, as will be shown in Sect. 3.4. Inside the wind farm, the wind speed is further reduced due to momentum extraction by the wind turbines. For the small wind farm, the wind speed decreases to $7.6\,\mathrm{m\,s^{-1}}$ at the wind farm trailing edge (TE). For the large wind farm, however, the wind speed reaches a minimum of $6.8\,\mathrm{m\,s^{-1}}$ approximately 40 km downstream of the leading edge (LE) and then increases again to $7.4\,\mathrm{m\,s^{-1}}$ at the wind farm TE. This acceleration is mainly caused by the large drop in the perturbation pressure of 30 Pa from the wind farm LE to TE. For the small wind farm this pressure drop is only approximately 7 Pa. The acceleration is also caused by the wind direction change and thus a greater ageostrophic wind speed component that results in a larger energy input by the geostrophic pressure gradient (Abkar and Porté-Agel, 2014). This will be shown in Sect. 3.5. In the wake of the large wind farm the wind speed increases further and reaches a maximum of $10.1\,\mathrm{m\,s^{-1}}$, which is 12 % more than the free-stream wind speed at the inflow. The maximum wind speed in the wake of the small wind farm exceeds the inflow wind speed by only 2 %. Further downstream the wind speed decreases again, indicating that it oscillates.

As shown in Fig. 3b, the wind direction is also significantly affected by the wind farms. Inside the wind farms the wind direction turns counterclockwise, reaching +2.3 and +10.1° at the TE of the small and large wind farm, respectively. Note that the wind direction already changes upstream of the wind farms, reaching +0.7 and +1.4° at the LE of the small and large wind farm, respectively. This wind direction change is caused by a reduction of the Coriolis force, which is a result of the reduced wind speed in and around the wind farms. For the large wind farm, the maximum deflection angle of 10.4° is reached inside the wind farm, at $x \approx 180\,\mathrm{km}$. Further downstream the wind turns clockwise, reaches $\Psi = 0°$ at $x \approx 330\,\mathrm{km}$ and turns further clockwise afterwards. For the small wind farm the maximum deflection angle of 2.8° is reached in the wake, at $x \approx 140\,\mathrm{km}$. The wind direction is zero at $x \approx 290\,\mathrm{km}$ and reaches a minimum at $x \approx 400\,\mathrm{km}$. Similar maximum deflection values of 2–3° have been reported in an LES study of Allaerts and Meyers (2016) for a 15 km long wind farm in conventionally neutral boundary layers.

The sinusoidal shape of the wind speed and wind direction evolution suggests that it is related to an inertial oscillation or an inertial wave along $x$. The wind direction has a +90° phase shift relative to the wind speed; i.e., the wind direction is zero where the wind speed has a maximum. The inertial wave has a wavelength of

$$\lambda_I \approx GT = 9.14\,\mathrm{m\,s^{-1}}\,14.6\,\mathrm{h} \approx 480\,\mathrm{km}, \tag{6}$$

where $G$ is the geostrophic wind speed and $T = 12\,\mathrm{h}/\sin(\phi) = 2\pi/f_3$ is the inertial period (Stull, 1988, p. 639). Consequently, the distance between wind direction maximum and minimum should be half a wavelength ($\lambda_I/2 = 240\,\mathrm{km}$), which corresponds well to the distance of 260 km that can be measured in the wake of the small wind farm. To add further confidence to this result, an additional simulation with a latitude of 80° N instead of 55° N is performed. The results are given in Appendix B and show that the wavelength decreases to $\lambda_I = 400\,\mathrm{km}$ due to the shorter inertial period at that latitude ($T = 12\,\mathrm{h}/\sin(80°) = 12.1\,\mathrm{h}$).

The inertial wave can also be seen in the hodograph of the hub height wind velocity components $u$ and $v$ along $x$,

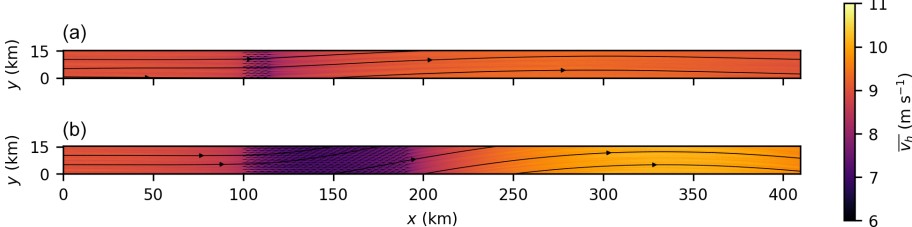

**Figure 2.** Mean horizontal wind speed at hub height for the small wind farm **(a)** and large wind farm **(b)**. Wind direction is indicated by streamlines.

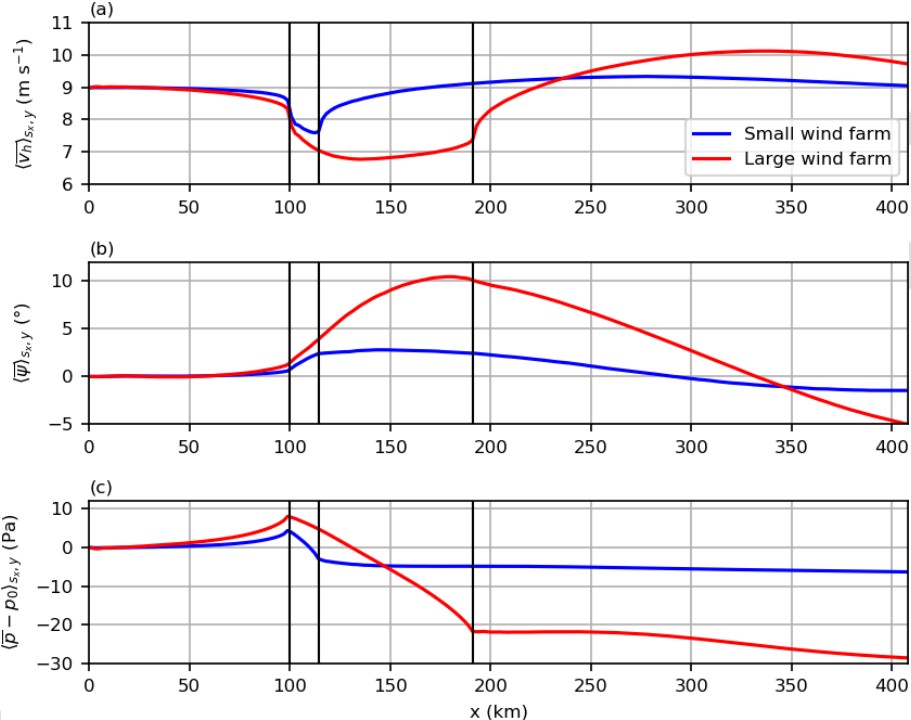

**Figure 3.** Horizontal wind speed **(a)**, wind direction **(b)** and perturbation pressure **(c)** along $x$. All quantities are averaged along $y$ and one turbine spacing along $x$. Vertical black lines indicate LE and TE of the small and large wind farm.

which is shown in Fig. 4. The figure shows that the oscillation is triggered by a reduction in $u$, followed by an increase in $v$. After the large perturbation by the wind farms, the hodograph approaches a circular path with clockwise direction. The center of these circular paths is the steady-state velocity of the inflow and not the geostrophic wind velocity. This is consistent with the findings of Baas et al. (2012), who investigated inertial oscillations in the nocturnal BL using the analytical model of van de Wiel et al. (2010) that accounts for frictional effects within the BL. The amplitude of the oscillations is $0.3\,\mathrm{m\,s^{-1}}$ for the small wind farm and $1.1\,\mathrm{m\,s^{-1}}$ for the large wind farm at $\lambda_I/4$ downstream of the respective wind farm trailing edge.

To investigate this effect in more detail, Fig. 5 shows the crosswise (perpendicular to streamlines) force components that act on the flow at hub height along $x$, averaged along $y$.

Shown are the vertical turbulent momentum flux divergence, the perturbation pressure gradient force and the geostrophic forcing (difference between geostrophic pressure gradient force and Coriolis force). Positive values indicate a counterclockwise deflection, and negative values indicate a clockwise deflection. The analysis is made from a Lagrangian frame of reference; thus, advection terms are not included. At the inflow all forces sum to zero and the mean flow is in a steady state. Due to the wind speed reduction upstream and inside the wind farms, the Coriolis force is reduced, so that the geostrophic pressure gradient force predominates and tends to deflect the flow counterclockwise. The vertical momentum flux divergence, however, tends to deflect the flow clockwise, but this force is weaker, so that the sum of these forces is still positive. Because the wind farms are infinite in the $y$ direction, the gravity waves are uniform in the

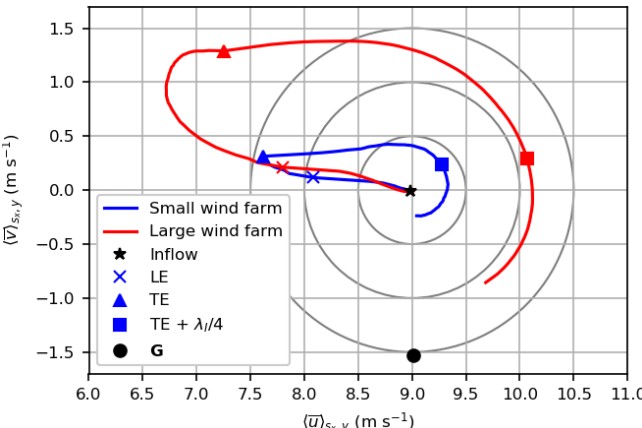

**Figure 4.** Hodographs of wind speed components $u$ and $v$ along $x$ at hub height. Special streamwise positions are marked: inflow, wind farm leading edge (LE), wind farm trailing edge (TE), one-quarter inertial wavelength downstream of the TE and the geostrophic wind velocity.

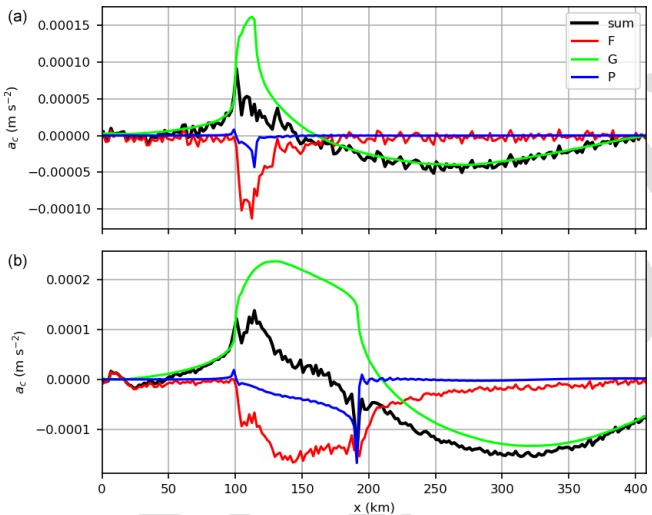

**Figure 5.** Crosswise forces (perpendicular to streamlines) at hub height along $x$, averaged along $y$ for the small wind farm **(a)** and large wind farm **(b)**. Shown are the divergence of the vertical turbulent momentum flux (resolved + SGS) $F$, the geostrophic forcing $G$ (difference between geostrophic pressure gradient force and Coriolis force) and the perturbation pressure gradient force $P$.

$y$ direction, and thus the perturbation pressure gradient force is parallel to the $x$ axis and has no effect on the wind direction at first. However, due to the change in wind direction further downstream inside the large wind farm, the perturbation pressure gradient force has a component perpendicular to the streamlines that tends to deflect the flow clockwise. At the end of the large wind farm the sum of all forces becomes negative, so that the flow begins to turn clockwise. Because the wind speed increases to super-geostrophic values in the wake, the Coriolis force becomes greater than the

geostrophic pressure gradient force so that the flow is deflected clockwise. The most significant difference between the small and the large wind farm is that the speed deficit in the large wind farm is greater and lasts longer. This results in a greater wind direction change and thus a greater inertial wave amplitude compared to the small wind farm. Whether a wind farm can trigger a significant inertial wave can be predicted by the Rossby number that relates inertia to Coriolis forces:

$$Ro = \frac{G}{L_{\mathrm{wf}} f_3}, \tag{7}$$

where $L_{\mathrm{wf}}$ is the length of the wind farm. An inertial wave occurs if the Rossby number has the order of magnitude of 1 or smaller. Coriolis effects become more dominant for smaller Rossby numbers so that the amplitude of the inertial wave is larger for the large wind farm ($Ro = 0.8$) than for the small wind farm ($Ro = 5.0$). That wind farms can trigger an inertial wave has not been reported by any other study, although there are studies that investigate wind farms with a similar size compared to the small wind farm in this study, e.g., Allaerts and Meyers (2016) or Wu and Porté-Agel (2017). The reason is that the inertial wave is more than 20 times longer than the small wind farm and is thus usually not covered by the numerical domain of other studies.

## 3.2 Turbulence at hub height

Figure 6 shows the total turbulence kinetic energy (TKE) and the turbulence intensity (TI) at hub height along $x$ for the small and large wind farm case. Both quantities are averaged along $y$ and piecewise averaged along $x$, where the averaging windows have a size of one turbine spacing and are centered between the turbine rows. The TKE and TI are calculated as follows:

$$\mathrm{TKE} = \frac{1}{2}(\overline{u'^2} + \overline{v'^2} + \overline{w'^2}) + \overline{e}, \tag{8}$$

$$\mathrm{TI} = \frac{\sqrt{\frac{2}{3}\mathrm{TKE}}}{\overline{v_h}}, \tag{9}$$

where an overbar indicates a temporal average; a prime indicates the deviation from this average; $\overline{u'^2}$, $\overline{v'^2}$ and $\overline{w'^2}$ are resolved-scale variances; and $\overline{e}$ is the SGS TKE. Upstream of the wind farms the ambient TKE is $0.22\,\mathrm{m^2\,s^{-2}}$. Within four turbine rows the TKE reaches a plateau value of $0.85\,\mathrm{m^2\,s^{-2}}$ for the small wind farm and $0.80\,\mathrm{m^2\,s^{-2}}$ for the large wind farm. The TKE is greater in the small wind farm because the wind speed is greater, and thus the turbines generate more TKE (see Fig. 3a). In the large wind farm the TKE decreases slightly to $0.76\,\mathrm{m^2\,s^{-2}}$ at the point where the minimum wind speed occurs. Further downstream the TKE increases to its maximum value of $0.85\,\mathrm{m^2\,s^{-2}}$ at the TE.

The TI shows a slightly different behavior than the TKE. Due to the normalization by the wind speed, which decreases

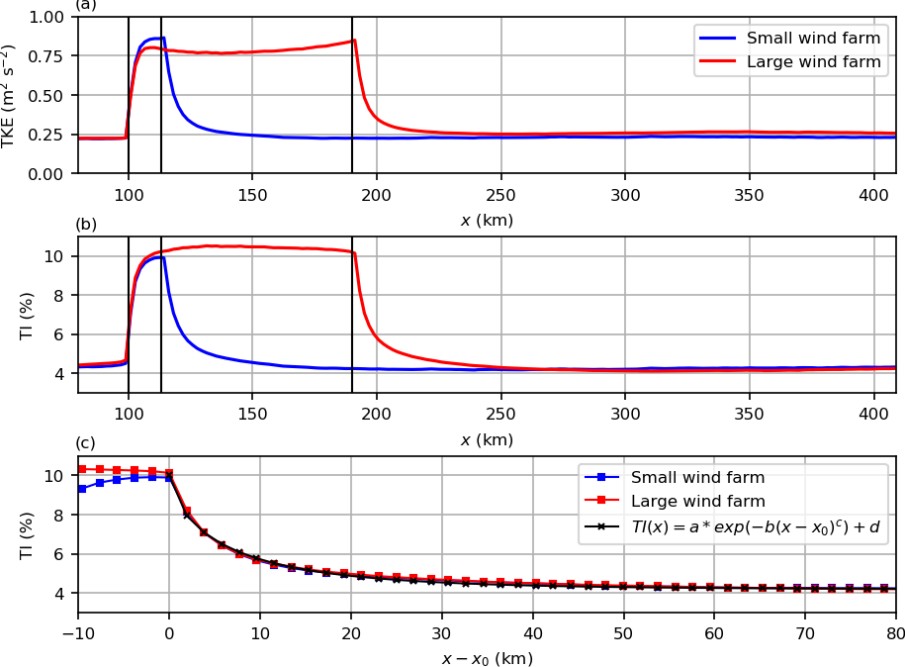

**Figure 6. (a)** Turbulence kinetic energy (TKE) and **(b, c)** turbulence intensity (TI) for the small and large wind farm case. The quantities are averaged along $y$ and piecewise along $x$, with averaging windows centered between the turbine rows. Vertical black lines indicate the wind farm LE and TE of the small and large wind farm. In panel **(c)** the graphs are shifted by $x_0$ so that the TE of the small and the large wind farm coincide, where $x_0 = x_{TE} + 0.5$ s. Parameters of the fitted curve are $a = 5.8\%$, $b = 0.28\,\mathrm{km}^{-c}$, $c = 0.68$ and $d = 4.2\%$.

upstream of the wind farms, the TI increases upstream of the wind farms. It increases from the ambient TI of 4.4 % to 4.6 % half a turbine spacing upstream of the LE. In the small wind farm, the TI reaches a plateau value of 9.9 %. In the large wind farm the TI is greater due to the smaller wind speed and reaches a maximum value of 10.5 % at the point where the minimum wind speed occurs. Further downstream, the TI decreases and reaches 10.1 % at the TE.

To compare the decay of the TI in the wake of the wind farms, the graphs in Fig. 6c are shifted so that the TEs of both wind farms coincide. It is remarkable that the decay of the TI in the wake of the small and the large wind farm follows exactly the same curve. This curve can be approximated by the following exponential function:

$$\mathrm{TI}(x) = a \exp(-b(x - x_0)^c) + d, \tag{10}$$

with coefficients $a = 5.8\%$, $b = 0.28\,\mathrm{km}^{-c}$, $c = 0.68$ and $d = 4.2\%$. Consequently, the wind farm size has no effect on the decay of the TI in wind farm wakes.

Further downstream, the TKE and the TI also show a slight oscillation as the wind speed and direction show (see Fig. 6a and b). However, the amplitude is much smaller than the TKE and TI levels that occur inside the wind farms, and thus the oscillations are hardly visible.

## 3.3 Boundary layer modification

The previous two sections focused on the flow at hub height. In this section it is shown how the wind farms modify the height and the internal structure of the BL.

The CBL is capped by an inversion layer (IL), which is displaced upwards due to the presence of the wind farms. The IL displacement $\delta$ is defined as the IL height $z_i$ relative to the IL height at the inflow:

$$\delta(x) = z_i(x) - z_i(x = 0), \tag{11}$$

where $z_i$ is defined as the height where the maximum vertical potential temperature gradient occurs. The IL displacement along $x$ is shown in Fig. 7 for the small and large wind farm case. The IL displacement begins already upstream of the wind farms and reaches +30 and +50 m at the LE of the small and large wind farm, respectively. Note that these changes in IL height (+5 % and +8 %) correspond well to the change in hub height wind speed (−5 % and −8 %; see Fig. 3) at the LE. This confirms that the IL displacement is a reaction of the flow to the speed reduction inside the boundary layer that ensures a constant mass flux inside the boundary layer. This has also been stated by other studies (Allaerts and Meyers, 2017; Maas and Raasch, 2022).

The maximum displacement is +55 m for the small wind farm and occurs near its TE. For the large wind farm the maximum displacement is +110 m and occurs approximately

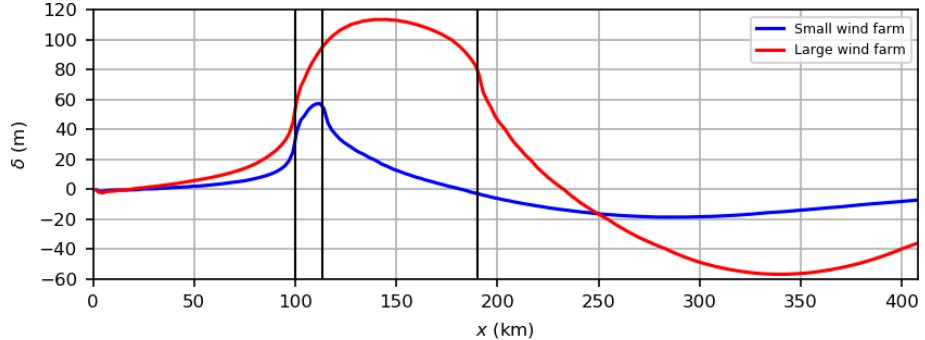

**Figure 7.** Inversion layer displacement $\delta$. Vertical black lines indicate LE and TE of the small and large wind farm.

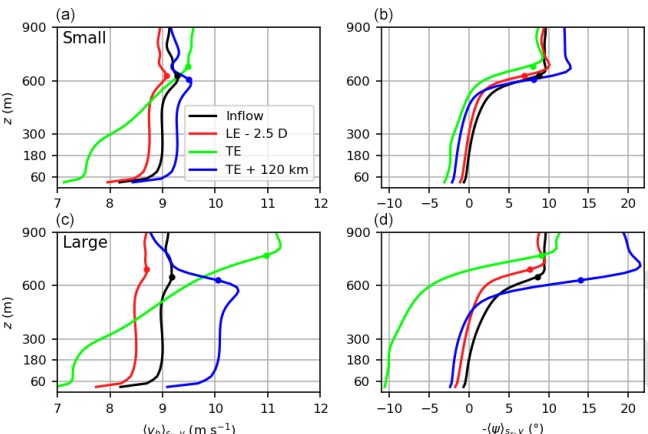

**Figure 8.** Vertical profiles of the horizontal wind speed **(a, c)** and the wind direction **(b, d)** for the small wind farm **(a, b)** and the large wind farm **(c, d)**. The profiles are averaged in time, along $y$ and over one turbine spacing along $x$. Horizontal lines are shown at $z = 60$ m (rotor bottom), $z = 180$ m (hub height), $z = 300$ m (rotor top) and $z = z_i = 600$ m (BL height at the inflow). Dots mark the BL height at the respective $x$ location.

40 km downstream of the LE. Thus, the maximum displacements occur at the location of the minimum wind speed (see Fig. 3). In the wake of the wind farms the IL displacement becomes negative, due to the increasing wind speed inside the boundary layer. For the small wind farm, the minimum displacement is $\delta \approx -20$ m and occurs at $x \approx 290$ km, corresponding to the location at which the hub height wind speed has a maximum and the wind direction is zero. The same holds for the large wind farm, except that the minimum displacement is $\delta \approx -55$ m and occurs at $x \approx 330$ km.

Besides the top of the BL, the internal structure of the BL is also significantly modified by the wind farms. Figure 8 shows vertical profiles of the wind speed and direction at several streamwise positions to demonstrate the development of the BL. As a reference, the inflow profiles are also shown. The second profile is located $2.5\,D$ upstream of the wind farm LEs. It shows that the speed deficit, caused by the block-

age effect, does not only occur at hub height but is rather constant over the entire BL. This is plausible because the speed reduction is caused by a positive streamwise pressure gradient, which is approximately constant over the entire height of the BL. At the wind farm TE, the wind speed at hub height is significantly reduced. At the BL top, however, the wind speed has increased from 9.0 to 9.6 m s$^{-1}$ for the small wind farm and from 8.6 to 11.0 m s$^{-1}$ for the large wind farm. Because turbulent momentum exchange is negligible at that height, these speed differences are solely caused by a drop in the perturbation pressure. The drop in the perturbation pressure between these points is 7 Pa for the small wind farm and 28 Pa for the large wind farm (see Fig. 3). Based on these pressure differences, Bernoulli's equation predicts these wind speed changes:

$$v_2 = \sqrt{\frac{2}{\rho}(p_1 - p_2) + u_1^2}$$

$$= \sqrt{\frac{2}{1.17\,\text{kg m}^{-3}}7\,\text{Pa} + (9.0\,\text{m s}^{-1})^2} = 9.6\,\text{m s}^{-1}$$

(small wind farm)

$$= \sqrt{\frac{2}{1.17\,\text{kg m}^{-3}}28\,\text{Pa} + (8.6\,\text{m s}^{-1})^2} = 11.0\,\text{m s}^{-1}$$

(large wind farm), (12)

which correspond to the observed wind speed changes. The pressure distribution in the BL is determined by gravity waves in the free atmosphere that are described in the next section. In the far wake, one-quarter of the inertial wavelength ($\lambda_I/4 = 120$ km) downstream of the wind farm TEs, the wind speed in the bulk of the BL is supergeostrophic. At 300 m height the wind speed has increased to 9.2 and 10.1 m s$^{-1}$ for the small and large wind farm, respectively.

The wind direction profiles of the small wind farm case show only small deviations of maximum $\pm 3°$ relative to the inflow profile. For the large wind farm case, however, the deviations can be as large as $\pm 10°$. Because the profiles of the

small and large wind farm case are qualitatively the same, only the large wind farm case is described in the following. At a distance of 2.5 $D$ upstream of the large wind farm, the wind direction has turned to the left by 1.4° at hub height and by 3.2° near the BL top. At the TE the wind direction has turned to the left by 10.0° up to a height of $\approx 600$ m. At the BL top the wind direction change is zero. One-quarter inertial wavelength downstream of the TE, the shape of the wind direction profile is nearly unchanged, but the wind direction has turned back to the right by approximately 8° relative to the profile at the TE. This also holds for the wind direction above the BL, indicating that there is also an inertial wave in the free atmosphere. This effect will be investigated in the next section.

## 3.4 Gravity waves

The displacement of the IL represents an obstacle for the flow in the overlying stably stratified free atmosphere and thus triggers atmospheric gravity waves. The gravity waves are investigated in more detail in this section because they induce streamwise pressure gradients at the surface and thus also affect the flow inside the BL and the energy budgets in the wind farms. Due to the large horizontal scales involved, Coriolis effects also affect the flow, so that the triggered gravity waves are not pure gravity waves but rather inertial gravity waves.

Figure 9 shows vertical cross sections of the horizontal wind speed and direction, the vertical wind speed, the perturbation pressure, and the potential temperature. The respective inflow profile is subtracted from each quantity, so that only the deviations from the steady-state mean flow remain. All quantities are averaged in time and along $y$. The different quantities show the expected pattern for stationary inertial gravity waves with upwards propagating energy; i.e., the phase lines are inclined upstream relative to the vertical. The phase relations between the quantities also correspond to the expected relations for gravity waves; e.g., $p$ and $w$ are in phase and $w$ and $\theta$ are 90° out of phase (Durran, 1990, Fig. 4.1).

The shown wave fields are a superposition of waves with three different inclination angles $\alpha$ (see Table 1). The first type of waves occurs above the wind farm LE and TE and is only visible in the vertical velocity field (see Fig. 9e and f). The phase lines are inclined by $\alpha_1 = 60°$ relative to the vertical. They are only visible in the vertical velocity field because the oscillation direction is much more vertical than that of the other wave types. The second type of waves appears above the wind farm with phase lines inclined by $\alpha_{2s} = 83.7$ and $\alpha_{2l} = 88.3°$ for the small and large wind farm, respectively. The third type of waves occurs above the wake and has phase lines that are inclined by $\alpha_3 = 89.3°$ (see dashed lines in Fig. 9a and b). The occurrence of these three different wave types can be explained by the shape of the topography, which is in this case the inversion layer. The wave type one is triggered by the sharp increase and decrease in IL height at the wind farm LE and TE (see Fig. 7). Wave types two and three, however, are triggered by the entire hill-like-shaped IL above the wind farm and the valley-like-shaped IL above the wake. The phase lines of wave type two are not perfectly straight but have a slightly positive curvature. The reason might be that the shape of the IL above the wind farm is not sinusoidal but is rather flat. The curved phase lines may also explain why the pressure distribution in the wind farm is not sinusoidal (as one could expect) but nearly linear (which is also true in the FA above the wind farm).

The amplitude of wave type one is approximately the same for the small and large wind farm case, while the amplitudes of wave types two and three are approximately 2 times greater for the large wind farm case relative to the small wind farm case (see Fig. 9 and note the different color scale ranges). The reason is that the IL displacement is twice as large for the large wind farm than for the small wind farm (see Fig. 7).

The wavelengths of the three different wave types are significantly different. For stationary waves, the horizontal wavelength can be calculated as the distance that an air parcel moves with the background velocity $U = U_{\mathrm{g}}$ during one oscillation period with oscillation frequency $\omega$:

$$\lambda_x = \frac{2\pi}{\omega} U. \tag{13}$$

The oscillation frequency $\omega$ of an inertial gravity wave is given by the dispersion relation (Pedlosky, 2003, Eq. 11.33):

$$\omega = \sqrt{f^2 \sin^2\alpha + N^2 \cos^2\alpha}, \tag{14}$$

where $N = \sqrt{\frac{g}{\theta_0}\Gamma} = 10.7 \times 10^{-3}\,\mathrm{s}^{-1}$ is the Brunt–Väisälä frequency. Note that the oscillation frequency is higher than for pure gravity waves because the Coriolis force acts as an additional restoring force. Equation (14) reduces to $\omega = N$ for pure vertical oscillating gravity waves (vertical phase lines) and to $\omega = f$ for pure horizontal oscillating inertial waves (horizontal phase lines). The absolute wavelength $\lambda$, i.e., the wavelength in the direction of phase propagation, is then given by

$$\lambda = \frac{2\pi c}{\omega} = \frac{2\pi U \cos\alpha}{\sqrt{f^2\sin^2\alpha + N^2\cos^2\alpha}} = \frac{1}{\sqrt{1 + \frac{f^2\sin^2\alpha}{N^2\cos^2\alpha}}} \frac{2\pi U}{N}, \tag{15}$$

so that the absolute wavelength becomes smaller for a larger $\alpha$. Note that for pure gravity waves, where the effect of $f$ can be neglected, the absolute wavelength is independent of $\alpha$ and corresponds to the Scorer length $L_{\mathrm{s}} = 2\pi U/N = 5.3$ km. The vertical wavelength is given by

$$\lambda_z = \frac{\lambda}{\sin\alpha}. \tag{16}$$

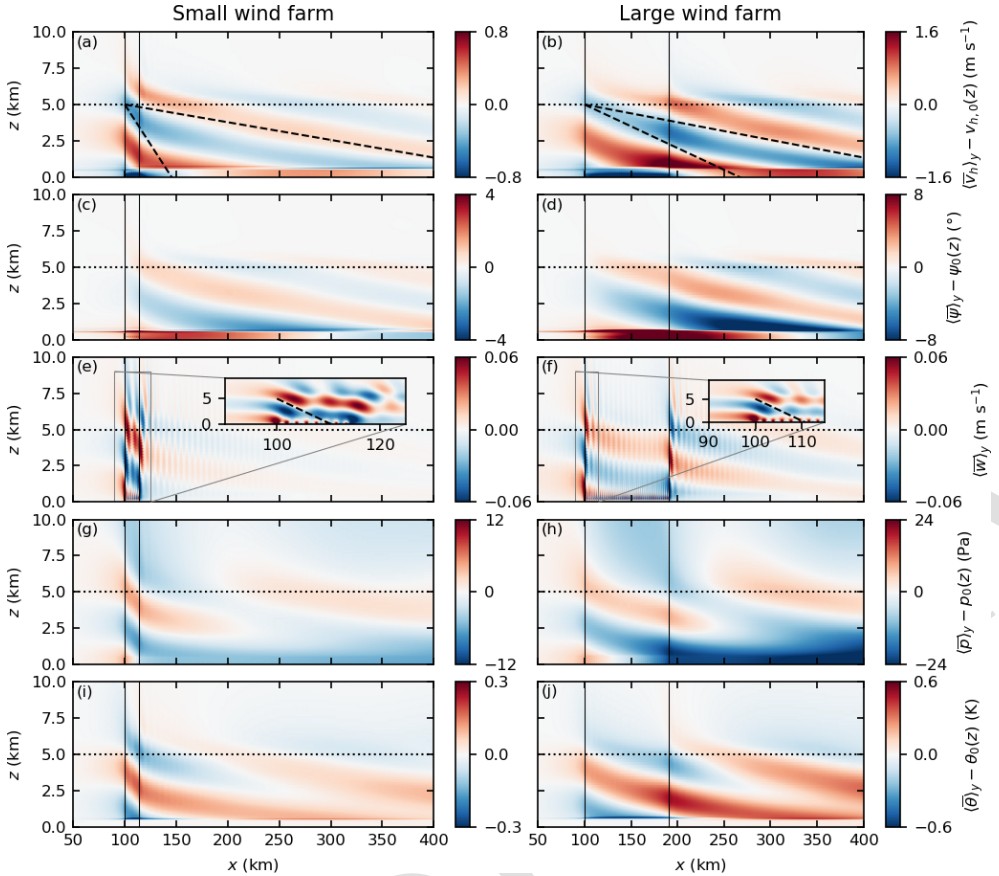

**Figure 9.** Vertical cross sections of the horizontal wind speed **(a, b)**, wind direction **(c, d)**, vertical wind speed **(e, f)**, perturbation pressure **(g, h)** and potential temperature **(i, j)**. The quantities are averaged in time and along $y$ and the respective mean inflow is subtracted. Vertical lines mark the leading and trailing edge of the small (left) and large (right) wind farm. The bottom of the Rayleigh-damping layer is marked by a dotted line. Dashed lines indicate the inclination angles of the phase lines. The gravity waves at the leading edges are shown in detail in panels **(e)** and **(f)**. Note that the range of the color scale is only half as large for the small wind farm than for the large wind farm except for $w$.

**Table 1.** Inclination angle of phase lines $\alpha$ and corresponding wavelengths for the different wave types present in Fig. 9.

| Wave type | $\alpha$ ° | $\lambda$ km | $\lambda_x$ km | $\lambda_z$ km | $\lambda_x/L_{wf}$ — | $L_{wf}/L_s$ — |
|---|---|---|---|---|---|---|
| 1 (LE + TE) | 60 | 5.29 | 10.6 | 6.11 | — | — |
| 2 (small wind farm) | 83.7 | 5.27 | 48.0 | 5.30 | 3.6 | 2.5 |
| 2 (large wind farm) | 88.3 | 4.96 | 167.1 | 4.96 | 1.85 | 17.0 |
| 3 (wake) | 89.3 | 3.91 | 320.3 | 3.91 | — | — |

The inclination angles of each wave type are measured in a figure that is similar to Fig. 9 but uses equal scales for both axes (not shown). The calculated oscillation frequencies and wavelengths of the three wave types are listed in Table 1.

The waves of type one have the smallest wavelength (10.6 km). Their effect on the pressure and horizontal velocity field is negligible. The horizontal wavelengths of wave type two are 48 and 167 km for the small and large wind farm, respectively. Why do these wavelengths occur? The ra-

tio of horizontal wavelength to the wind farm length is 3.6 for the small wind farm and 1.85 for the large wind farm, so that the wind farm length is not a good measure to explain the wavelength. But the wavelength can be explained by the shape of the IL. The horizontal distance between the largest slope of the IL (at the LE) and the location of the maximum displacement is 12 and 42 km for the small and large wind farm, respectively (see Fig. 7). These distances correspond very well to $\lambda_{x,2}/4$ of the waves above the wind farm.

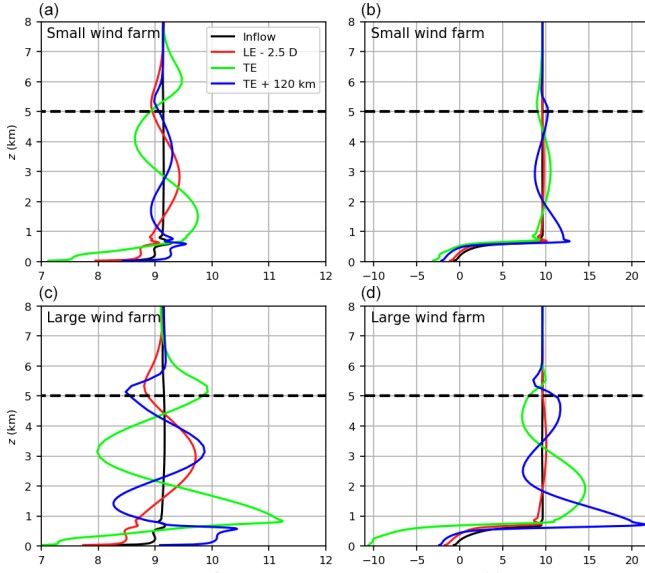

**Figure 10.** Vertical profiles of the horizontal wind speed **(a, c)** and the wind direction **(b, d)** for the small wind farm **(a, b)** and the large wind farm **(c, d)**. The profiles are averaged in time, along $y$ and over one turbine spacing along $x$. Horizontal dashed line indicates the bottom of the Rayleigh damping layer.

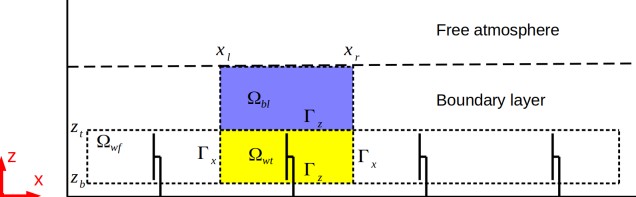

**Figure 11.** Sketch of wind turbine control volumes $\Omega_{wt}$, BL control volumes $\Omega_{bl}$ and the wind farm control volume $\Omega_{wf}$. In the $x$ direction the control volumes are bounded by the surfaces $\Gamma_x$ at $x = x_l$ and $x = x_r$. In the vertical direction, the control volumes are bounded by $\Gamma_z$ at $z = z_b$ and $z = z_t$. In the $y$ direction the control volumes are bounded by the cyclic domain boundaries. The control volumes are centered on the respective turbine hub.

The presented results generally correspond very well to the results of Allaerts and Meyers (2017), who investigated gravity waves above a 15 km long wind farm, which approximately corresponds to the length of the small wind farm in this study. One significant difference between the studies is the larger extent of the large wind farm in this study, causing inertial gravity waves due to Coriolis effects that become dominant at that scale. The second significant difference is the weaker stratification of $+1.0\,\mathrm{K\,km^{-1}}$ in their study compared to $+3.5\,\mathrm{K\,km^{-1}}$ in this study. This leads to a different Brunt–Väisälä frequency and thus a different Scorer length (which corresponds to the absolute wavelength of stationary pure gravity waves). Consequently, the wind farm in Allaerts and Meyers (2017) has approximately the length of the Scorer length ($L_{wf}/L_s = 15\,\mathrm{km}/12.8\,\mathrm{km} \approx 1.2$), whereas the small wind farm and large wind farm in this study are $L_{wf,s}/L_s = 13.44\,\mathrm{km}/5.3\,\mathrm{km} \approx 2.5$ and $L_{wf,l}/L_s = 90.24\,\mathrm{km}/5.3\,\mathrm{km} \approx 17.0$ times longer than the Scorer length, respectively. Due to the large ratio of $L_{wf}/L_s$ in the large wind farm case, the waves at the wind farm LE and TE (type one) are separated by several wavelengths and can thus be clearly distinguished from wave type two in this study. However, the less orderly shape of the $w$ field in Allaerts and Meyers (2017) (their Fig. 12b) suggests that wave type one is also present there.

The vertical structure of the gravity waves is shown by profiles of the wind speed and wind direction at different streamwise positions in Fig. 10. It can be seen that the amplitude of the waves is approximately twice as large in the large wind farm case than in the small wind farm case, as already mentioned above. There is a phase shift of approximately 90° between the wind speed and wind direction. Inside the Rayleigh damping layer the wind speed variations decay within 3 km, and the wind direction variations decay within 1 km.

## 3.5 Energy budget analysis

Wind turbines extract kinetic energy from the BL flow and convert it into electrical energy. Consequently, there is less energy available for wind turbines located in the wake of upstream wind turbines. The energy extraction is considered by a velocity deficit zone in the wind turbine wake in classical wake models such as Jensen (1983). However, there are also sources of energy that add new kinetic energy into the BL. As will be shown in this section, these sources depend on the above-discussed flow effects and significantly affect the wind turbine power, especially for the large wind farm.

To analyze the different energy sources and sinks in the BL, an extensive energy budget analysis is presented in this section. The analysis is very similar to the energy budget analysis made by Allaerts and Meyers (2017) for a 15 km long wind farm. The energy budgets are calculated for three different control volumes. The control volume $\Omega_{wt}$ envelops the wind turbine rotor, the control volume $\Omega_{bl}$ envelops the rest of the BL above $\Omega_{wt}$ and the entire wind farm is enveloped by control volume $\Omega_{wf}$, which is the sum of all $\Omega_{wt}$ (see Fig. 11). The control volumes have a streamwise length of one turbine spacing and are centered at the respective wind turbine hub. The bottom and top boundaries of $\Omega_{wt}$ are $(z_b, z_t) = (50, 310)\,\mathrm{m}$, which is $1\mathrm{d}z$ larger than the rotor diameter to cover the smeared forces of the wind turbine model. The bottom and top boundaries of $\Omega_{bl}$ are $(z_b, z_t) = (310\,\mathrm{m}, z_i(x))$. In the $y$ direction the control volumes are bounded by the cyclic domain boundaries.

The equation for the conservation of the resolved-scale kinetic energy can be obtained by multiplying PALM's momentum equation (Eq. 2) with $u_i$, averaging in time, assum-

ing stationarity and integrating over the control volume $\Omega$:

$$
\begin{aligned}
0 = & \underbrace{-\int_\Omega \frac{\partial \overline{\tilde{u}}_j \overline{E}_k}{\partial x_j} d\Omega}_{\mathcal{A}} \underbrace{-\int_\Omega \frac{\overline{\tilde{u}}_i}{\rho_0} \frac{\partial \overline{\pi^*}}{\partial x_i} d\Omega}_{\mathcal{P}} \\
& \underbrace{-\int_\Omega \frac{\partial}{\partial x_j} \overline{\tilde{u}}_i \overline{\tilde{u}'_i \tilde{u}'_j} d\Omega + \int_\Omega \frac{\partial}{\partial x_j} \overline{\tilde{u}_i \tau_{ij}} d\Omega}_{} \\
& \underbrace{-\int_\Omega \frac{\partial}{\partial x_j} \frac{1}{2} \overline{\tilde{u}'_j \tilde{u}'_i \tilde{u}'_i} d\Omega - \int_\Omega \frac{\overline{\tilde{u}'_i}}{\rho_0} \frac{\partial \pi^{*\prime}}{\partial x_i} d\Omega}_{\mathcal{F}} \\
& \underbrace{+\int_\Omega (\overline{\tilde{u}}_2 f_3 u_{g,1} - \overline{\tilde{u}}_1 f_3 u_{g,2}) d\Omega}_{\mathcal{G}} \\
& \underbrace{+\int_\Omega \frac{g}{\theta_0} \overline{(\tilde{\theta} - \theta_0) \tilde{u}_3} d\Omega}_{\mathcal{B}} \\
& \underbrace{-\int_\Omega \overline{\tau_{ij} \frac{\partial \tilde{u}_i}{\partial x_j}} d\Omega - \mathcal{R}}_{\mathcal{D}} + \underbrace{\int_\Omega \overline{\tilde{u}_i d_i} d\Omega}_{\mathcal{W}}.
\end{aligned}
\tag{17}
$$

Note TS1 that the mean kinetic energy (KE, $\overline{E}_k$) contains the kinetic energy of the mean flow (KEM) and the turbulence 5 kinetic energy (TKE) of the resolved flow:

$$
\overline{E}_k = \frac{1}{2} \overline{\tilde{u}_i \tilde{u}_i} = \frac{1}{2} \overline{\tilde{u}}_i \overline{\tilde{u}}_i + \frac{1}{2} \overline{\tilde{u}'_i \tilde{u}'_i}.
\tag{18}
$$

The terms of Eq. (17) are categorized as follows:

- $\mathcal{A}$ is the divergence of KE advection;

- $\mathcal{P}$ is the energy input by mean perturbation pressure gra-
10   dients;

- $\mathcal{F}$ is the transport of KEM by resolved turbulent stresses (term 1), transport of KEM and TKE by SGS stresses (term 2), turbulent transport of resolved-scale TKE by velocity fluctuations (term 3), and turbulent transport of
15   KE by perturbation pressure fluctuations (term 4);

- $\mathcal{G}$ is the energy input by geostrophic forcing;

- $\mathcal{B}$ is the energy input by buoyancy forces;

- $\mathcal{D}$ is the dissipation by SGS model and residual $\mathcal{R}$; and

- $\mathcal{W}$ is the energy extraction by wind turbines.

20 Equation (17) has a positive residual $\mathcal{R}$ because the magnitude of the calculated dissipation is underestimated, which has two reasons. First, the local velocity gradients are underestimated because they are calculated with central differences. Second, the fifth-order upwind advection scheme

of Wicker and Skamarock (2002) has numerical dissipation, 25 suppressing the magnitude of the smallest eddies, for which the gradients and the dissipation are highest (Maronga et al., 2013). The residual is subtracted from the (negative) dissipation term $\mathcal{D}$ to compensate for the underestimated magnitude of the calculated dissipation. 30

Instead of calculating terms $\mathcal{A}$ and $\mathcal{F}$ as a volume integral, they can also be calculated as a surface integral over the control volume surfaces (Gauss's theorem):

$$
\mathcal{A} = \underbrace{\left[\int_{\Gamma_x} \left(-\overline{\tilde{u}}_1 \overline{E}_k\right) d\Gamma_x\right]_{x_l}^{x_r}}_{\mathcal{A}_x} + \underbrace{\left[\int_{\Gamma_z} \left(-\overline{\tilde{u}}_3 \overline{E}_k\right) d\Gamma_z\right]_{z_b}^{z_t}}_{\mathcal{A}_z},
\tag{19}
$$

$$
\mathcal{F} = \underbrace{\left[\int_{\Gamma_x} \left(-\overline{\tilde{u}}_i \overline{\tilde{u}'_i \tilde{u}'_1} + \overline{\tilde{u}_i \tau_{i1}} - \frac{1}{2} \overline{\tilde{u}'_1 \tilde{u}'_i \tilde{u}'_i} - \frac{\overline{\tilde{u}'_1 \pi^{*\prime}}}{\rho_0}\right) d\Gamma_x\right]_{x_l}^{x_r}}_{\mathcal{F}_x}
$$
$$
+ \underbrace{\left[\int_{\Gamma_z} \left(-\overline{\tilde{u}}_i \overline{\tilde{u}'_i \tilde{u}'_3} + \overline{\tilde{u}_i \tau_{i3}} - \frac{1}{2} \overline{\tilde{u}'_3 \tilde{u}'_i \tilde{u}'_i} - \frac{\overline{\tilde{u}'_3 \pi^{*\prime}}}{\rho_0}\right) d\Gamma_z\right]_{z_b}^{z_t}}_{\mathcal{F}_z},
\tag{20}
$$
35

where $\mathcal{A}_x$ and $\mathcal{A}_z$ are the advection of KE through the left-/right and bottom/top surfaces, respectively, and $\mathcal{F}_x$ and $\mathcal{F}_z$ are turbulent fluxes through the left/right and bottom/top surfaces, respectively.

### 3.5.1 Energy budgets for the entire small and large wind 40 farm

The energy budgets for a control volume that envelops the entire small/large wind farm are shown in Fig. 12. The budget terms of Eq. (17) are converted from $W\rho^{-1}$ to MW per turbine to make them more meaningful. The air density is 45 $\rho = 1.17\,\text{kg m}^{-3}$.

With 5.6 MW per turbine, the horizontal advection of kinetic energy ($\mathcal{A}_x$) is the greatest energy source for the small wind farm. For the large wind farm, however, this source is only as large as 0.9 MW per turbine. This large difference 50 is mainly the result of the fact that the large wind farm is 6 times longer than the small one, so that the influx of KE at the wind farm LE is distributed over 6 times more turbine rows. Additionally, the wind speed at the TE of the large wind farm is larger than at the TE of the small wind farm, so that more 55 KE leaves the wind farm control volume (see Figs. 3 and 13).

For both wind farms, approximately 40 % of $\mathcal{A}_x$ leaves the wind farm control volume again through vertical advection $\mathcal{A}_z$. KE is leaving the top of the control volume by a mean positive $w$, which is the result of the turbine-induced flow 60 deceleration and the requirement for mass flow conservation. This effect has also been described by Allaerts and Meyers (2017).

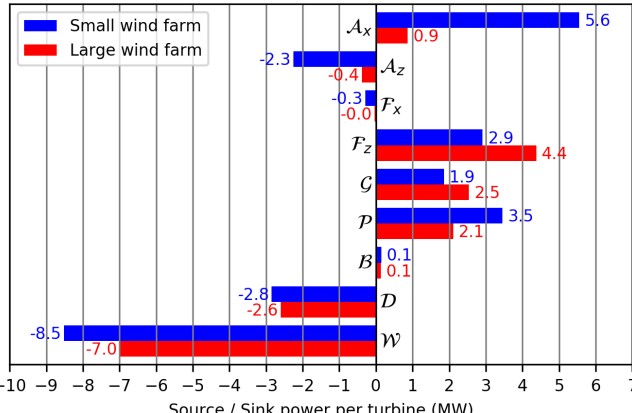

**Figure 12.** Energy budgets inside control volume $\Omega_{\mathrm{wf}}$ that envelops the entire small/large wind farm. The budget terms are horizontal advection of KE ($\mathcal{A}_x$), vertical advection of KE ($\mathcal{A}_z$), turbulent fluxes through left/right ($\mathcal{F}_x$) and bottom/top ($\mathcal{F}_z$) surfaces, geostrophic forcing ($\mathcal{G}$), perturbation pressure gradients ($\mathcal{P}$), buoyancy ($\mathcal{B}$), dissipation ($\mathcal{D}$), and wind turbines ($\mathcal{W}$).

The horizontal turbulent fluxes $\mathcal{F}_x$ are a small energy sink of $-0.3$ MW per turbine for the small wind farm. This sink is mainly caused by a net outflow of TKE in the first three turbine rows, where the incoming flow contains less TKE than the outgoing flow (see Figs. 6a and 13). For the large wind farm $\mathcal{F}_x$ is negligible because the described effect spreads over 6 times more turbine rows.

The vertical turbulent flux of KE ($\mathcal{F}_z$) is the greatest energy source for the large wind farm, contributing 4.4 MW per turbine. For the small wind farm it is the third largest energy source with 2.9 MW per turbine. These results show that for large wind farms the vertical turbulent flux of KE is much more important than the horizontal advection ($\mathcal{F}_z \approx 5 \times \mathcal{A}_x$), whereas for small wind farms the horizontal advection of KE is more important ($\mathcal{A}_x \approx 2 \times \mathcal{F}_z$).

The energy input by the geostrophic forcing ($\mathcal{G}$) is the fourth largest energy source for the small wind farm (1.9 MW per turbine) but the second largest energy source for the large wind farm (2.5 MW per turbine). The 32 % higher value for the large wind farm is the result of the wind direction change that is triggered by the wind farm itself (see Fig. 3). It causes the ageostrophic wind velocity component to rise and thus leads to a higher energy input (see also Figs. 13 and 14). This effect has also been shown for infinitely large wind farms by Abkar and Porté-Agel (2014) and finite, large wind farms by Maas and Raasch (2022).

The energy input by the mean perturbation pressure gradient ($\mathcal{P}$) is the second largest energy source for the small wind farm (3.5 MW per turbine) and the third largest energy source for the large wind farm (2.1 MW per turbine). For the large wind farm $\mathcal{P}$ is approximately 60 % of $\mathcal{P}$ for the small wind farm, although the difference in perturbation pressure between the LE and TE of the large wind farm is approx-

imately 4.3 times larger than that of the small wind farm (30 Pa / 7 Pa; see Fig. 3). However, this difference spreads over a 6 times longer wind farm, so that the resulting pressure gradient is only 70 % as large. The term $\mathcal{P}$ also depends on the mean wind speed, which is generally smaller in the large wind farm, resulting in a further reduction of $\mathcal{P}$.

The production of KE by buoyancy ($\mathcal{B}$) is negligibly small for the small and large wind farm case. This is an expected result for the offshore-typical weakly unstable CBL with $L \approx -400$ m. However, this term might be much larger for strong CBLs.

The total of all above named sources ($\mathcal{A} + \mathcal{F} + \mathcal{G} + \mathcal{P} + \mathcal{B}$) is 11.3 MW per turbine for the small wind farm and 9.6 MW per turbine for the large wind farm. For the small wind farm 75 % of this available power is used by the wind turbines ($\mathcal{W} = -8.5$ MW per turbine), and for the large wind farm it is 73 % ($\mathcal{W} = -7.0$ MW per turbine). The rest of the available energy is lost by dissipation ($\mathcal{D}$).

### 3.5.2 Energy budgets in the turbine control volumes

The energy budgets inside the wind turbine control volumes $\Omega_{\mathrm{wt}}$ are shown in Fig. 13. In the first two turbine rows the horizontal advection of KE ($\mathcal{A}_x$) is the dominant energy source. A large amount of this KE, however, is lost by vertical advection of KE through the control volume top. This effect is caused by the fact that any horizontal convergence (flow deceleration with positive $\mathcal{A}_x$) requires a vertical divergence (negative $\mathcal{A}_z$) so that the mass flux is conserved. Consequently, the shape of $\mathcal{A}_z$ is qualitatively the vertically mirrored shape of $\mathcal{A}_x$. At row 21 of the large wind farm the terms change sign because from there on the flow accelerates again (see Fig. 3). For the small wind farm this happens between the last two rows. From there on, more KE leaves the control volume than KE enters the control volume in the streamwise direction. But $\mathcal{A}_z$ is then positive, indicating that KE is transported into the wind farm from above by a negative mean vertical velocity. The flow acceleration at the end of the wind farms is mainly caused by the negative perturbation pressure gradient that has the highest magnitude at the wind farm TE (see Fig. 3). The energy input by the pressure gradient $\mathcal{P}$ thus increases towards the TE of the large wind farm and reaches 5 MW per turbine at the TE. The pressure distribution inside the wind farm is determined by wave type two of the gravity waves (see Sect. 3.4 and Fig. 9). The flow acceleration near the TE of the wind farm and the related negative net advection of KE have also been reported by Allaerts and Meyers (2017) for a 15 km long wind farm in a conventionally neutral BL.

The horizontal turbulent fluxes are a weak energy sink ($\approx -1$ MW per turbine) in the first two rows because the outgoing flow contains more TKE than the incoming flow.

For both wind farms the vertical turbulent fluxes are zero at the first row. For the small wind farm they rise from 3 MW in the middle of the wind farm to 4 MW at the TE. For the

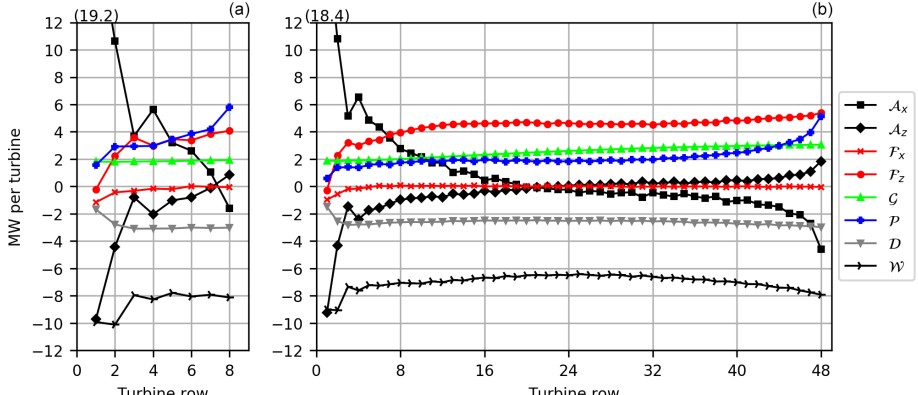

**Figure 13.** Energy budgets inside the wind turbine control volumes $\Omega_{\mathrm{wt}}$ for the small wind farm **(a)** and the large wind farm **(b)**. The budget terms are horizontal advection of KE ($\mathcal{A}_x$), vertical advection of KE ($\mathcal{A}_z$), turbulent fluxes through left/right ($\mathcal{F}_x$) and bottom/top ($\mathcal{F}_z$) surfaces, geostrophic forcing ($\mathcal{G}$), perturbation pressure gradients ($\mathcal{P}$), dissipation ($\mathcal{D}$), and wind turbines ($\mathcal{W}$). The buoyancy term ($\mathcal{B}$) is not shown because it is very small.

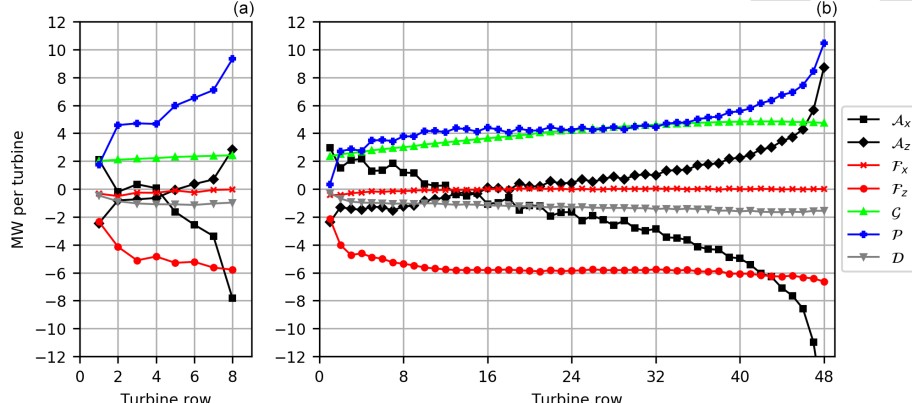

**Figure 14.** Energy budgets inside the BL control volumes $\Omega_{\mathrm{bl}}$ for the small wind farm **(a)** and the large wind farm **(b)**. The budget terms are horizontal advection of KE ($\mathcal{A}_x$), vertical advection of KE ($\mathcal{A}_z$), turbulent fluxes through left/right ($\mathcal{F}_x$) and bottom/top ($\mathcal{F}_z$) surfaces, geostrophic forcing ($\mathcal{G}$), perturbation pressure gradients ($\mathcal{P}$), and dissipation ($\mathcal{D}$). The buoyancy term ($\mathcal{B}$) is not shown because it is very small. There is no energy extraction by wind turbines ($\mathcal{W}$) in the BL control volume.

large wind farm they stay constantly at 4.5 MW per turbine from approximately row 14, but from row 32 they start to rise again, reaching 5.5 MW per turbine at the TE. The values of $\mathcal{F}_z$ are generally greater for the large wind farm because there is more energy available in the upper part of the BL, which is mainly the result of the higher energy input by the geostrophic forcing for the large wind farm (see Fig. 14). From row 7 to the TE of the large wind farm the vertical turbulent fluxes are the greatest energy source of all terms.

For the small wind farm, the energy input by the geostrophic forcing is approximately constant at 2 MW per turbine. For the large wind farm, however, it steadily rises from 2 MW per turbine at the LE to 3 MW per turbine at the TE. As described in the last section, this effect is caused by the wind direction change along the wind farm that leads to a higher ageostrophic wind velocity component.

The wind turbines in the first two rows of the small and large wind farm extract approximately 10.0 and 9.0 MW, respectively (remember the staggered turbine configuration). The wind turbine power is constant at 8.0 MW in the rest of the small wind farm. In the large wind farm, however, the turbine power slowly decreases to 6.5 MW at row 24 and then increases to nearly 8.0 MW at the last turbine row. This power increase is the result of the wind speed increase in the second half of the wind farm that is related to the wind direction change and increase in $\mathcal{G}$.

The energy dissipation is approximately constant at $\mathcal{D} = -3$ MW per turbine in the small wind farm and at $\mathcal{D} = -2.5$ MW in the large wind farm, except for the first 3 rows, where it is smaller. At the TE of the large wind farm $\mathcal{D}$ is slightly higher than in the middle, which can be related to the higher TKE at that location (see Fig. 6).

### 3.5.3 Energy budgets in the boundary layer control volumes

In the BL control volumes above the wind turbines the flow begins to accelerate earlier than inside the wind farm (row 4 of the small wind farm and row 14 of the large wind farm), as indicated by the evolution of $\mathcal{A}_x$ (see Fig. 14). The energy for this acceleration is provided by $\mathcal{G}$ and $\mathcal{P}$ in approximately equal parts (4 MW per turbine) in the large wind farm, except towards the TE, where $\mathcal{P}$ increases steeply due to a significant drop on perturbation pressure (see Fig. 3). For the small wind farm $\mathcal{P}$ is 2 to 4 times larger than $\mathcal{G}$, except at the first row, where they are equal.

In the small wind farm $\mathcal{G}$ increases by only 10 % from LE to TE, but in the large wind farm it increases by more than 100 % (from 2.2 to 4.8 MW per turbine). This is a much larger increase than in the wind turbine control volume, although the wind direction change is the same at all heights (see Fig.8). However, the wind speed is much greater above the wind farm, resulting in a higher ageostrophic wind velocity component and thus a higher $\mathcal{G}$.

The vertical turbulent fluxes $\mathcal{F}_z$ have the same shape as but opposite sign to the turbine control volumes (see Fig. 13) because they transfer energy from the BL down into the wind farm. Their magnitude is approximately 25 % smaller in the turbine control volume than in the BL control volume because there is also a KE loss through the bottom of the turbine control volumes.

## 4   Conclusions

The aim of this LES study is to provide a systematic comparison between small and large wind farms, focusing on the flow effects and the energy budgets in and around the wind farms. The size of the wind farms is chosen to be representative for current wind farm clusters (length of approximately 15 km) and future wind farm clusters (length of approximately 90 km).

The results show that there are significant differences in the flow field and the energy budgets of the small and large wind farm. The large wind farm triggers an inertial wave with a wind direction amplitude of approximately 10° and a wind speed amplitude of more than $1 \, \text{m s}^{-1}$. In a certain region in the far wake of a large wind farm the wind speed is greater than far upstream of the wind farm. The inertial wave also exists for the small wind farm, but the amplitudes are approximately 4 times weaker and thus may be hardly observable in real wind farm flows that are more heterogeneous. The decay of turbulence intensity in the wind farm wakes follows an exponential function and does not depend on the wind farm length. Thus, regarding turbulence, the wake of large wind farms has the same length as that of small wind farms. The wind-farm-induced speed deficit causes an upward displacement of the IL, triggering inertial gravity waves above the small and large wind farm. Because the inertial gravity waves

have a substantial effect on the energy budgets in the wind farm, their existence should be proven by measurements in the future. However, this might be a difficult task because the amplitudes in the vertical wind speed and pressure are very small ($0.05 \, \text{m s}^{-1}$ and 20 Pa).

The energy budget analysis shows that the dominant energy source in small wind farms is the advection of kinetic energy. For large wind farms, however, the advection is much less important and the energy input by vertical turbulent fluxes becomes dominant. Due to the wind-farm-induced wind direction change and the related increase in the ageostrophic wind speed, the energy input by the geostrophic forcing (synoptic-scale pressure gradient) can increase by more than 100 %. This result shows that the presence of large offshore wind farm clusters will modify the offshore, low-roughness BL towards a more onshore-typical, high-roughness BL. This leads to a faster wake recovery and allows for smaller turbine spacings. The energy budget analysis shows that the power output of large wind farms depends on several different energy sources that are determined by the flow state inside and above the BL. Simple wake models do not take these different sources into account and are expected to be inappropriate for accurate power predictions of large wind farms. Proving this hypothesis is an open research tasks.

The results in this study are based on very idealized simulation setups, assuming a homogeneous surface and a barotropic flow with constant geostrophic wind over a horizontal distance of 400 km and a constant lapse rate over a vertical distance of 5 km. These idealized conditions rarely occur in reality. A deviation from these idealized conditions could distort and weaken the described effects. Additionally, only one meteorological setup is used in this study. A change in BL height, stability or wind speed may affect the results significantly. Consequently, the presented results are a first qualitative guess of what is different in large wind farms compared to small wind farms. Further research is needed to find out how sensitive the results are to the named assumptions and to changes in the meteorological conditions and the turbine spacing. The largest deviation from reality is probably introduced by the assumption of an infinitely wide wind farm. The investigation of a multi-gigawatt wind farm with a finite size in both lateral directions will be the subject of a follow-up study.

## Appendix A: Validity of the Boussinesq approximation

The domain height in this study is much larger than in most large-eddy simulation studies that mainly cover the boundary layer. The incompressibility assumption requires the involved vertical length scales to be much smaller than $c^2/g \approx$ 12 km, where $c$ is the speed of sound (Stull, 1988, p. 77). Therefore, the question of whether the Boussinesq approxi-

mation that assumes a constant density is still valid for these simulations arises. To clarify this question, two additional test simulations were performed. One using the Boussinesq approximation and the other using the anelastic approxima-
5  tion, for which the density can vary with height. The results are shown in Figs. A1–A3. The gravity waves are qualitatively the same in both cases (wavelength, angles of the phase lines). But there are some quantitative differences at greater heights (e.g., 8 km). At that height, the velocity and temper-
10  ature amplitudes are greater and the pressure amplitudes are smaller for the anelastic approximation. But these differences do not affect the results at hub height (wind speed, direction and perturbation pressure). Therefore, it is appropriate to use the Boussinesq approximation for the simulations in this study.

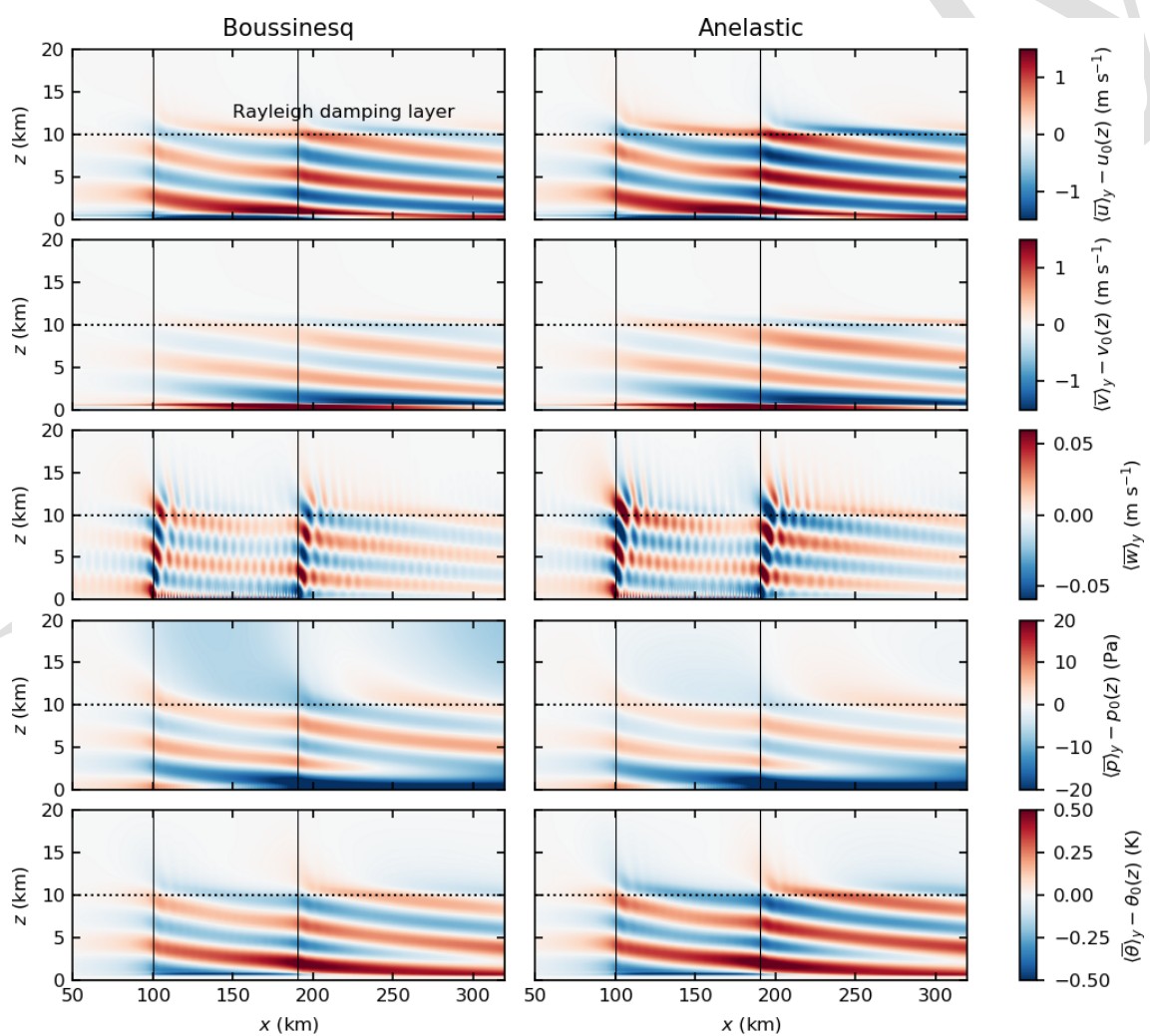

**Figure A1.** Wave fields for the test simulation with Boussinesq approximation (left) and anelastic approximation (right). All quantities are averaged in time and along $y$ and are given as deviations to the inflow profile.

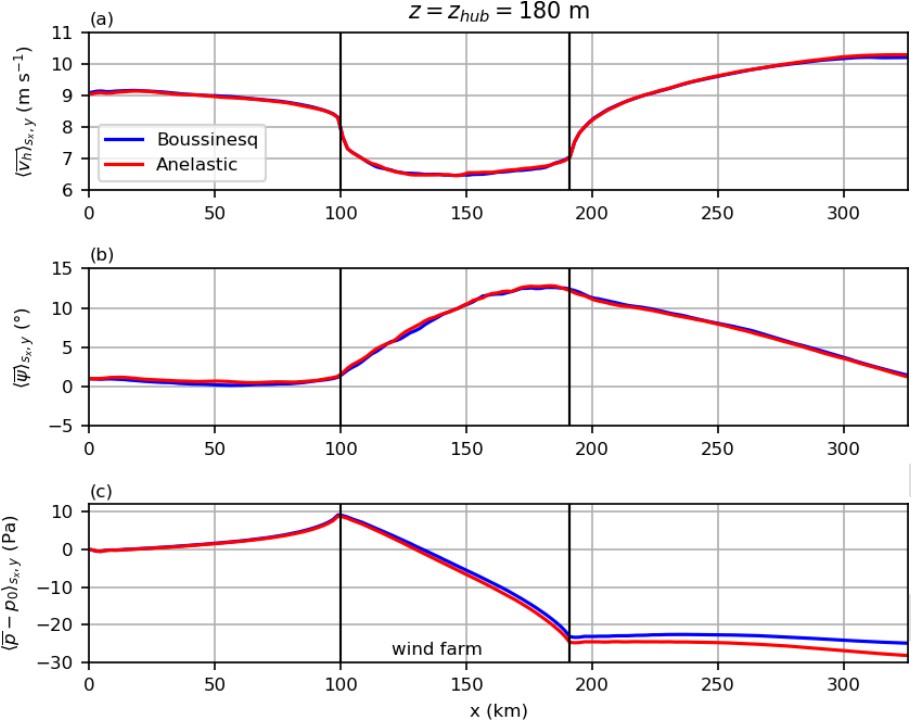

**Figure A2.** Horizontal wind speed, wind direction and relative perturbation pressure at hub height for both approximation types.

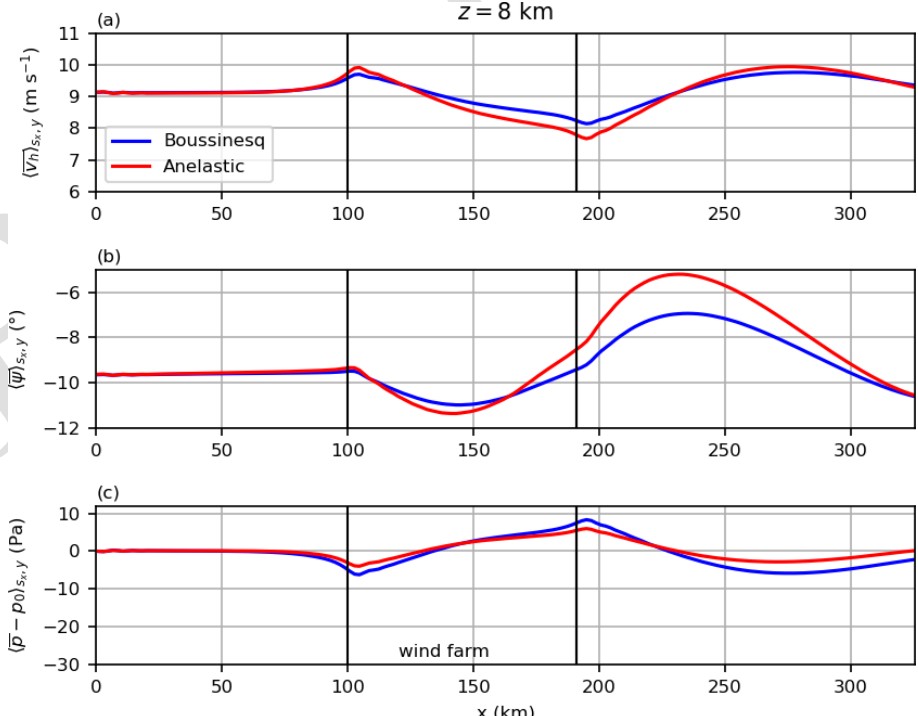

**Figure A3.** Horizontal wind speed, wind direction and relative perturbation pressure at 8 km height for both approximation types.

## Appendix B: Simulation with different latitude

Two additional large wind farm simulations with two different latitudes are performed to prove that the observed wave in the wake is an inertial wave. The domain length is increased further to 655.36 km to capture approximately one wavelength. The latitudes $\phi_1 = 55°$ (original simulation) and $\phi_2 = 80°$ (additional simulation) are used. The larger latitude should result in a shorter inertial period ($T = 12\,\text{h}/\sin(80°) = 12.1\,\text{h}$) and thus a shorter wavelength ($\lambda_I \approx GT \approx 400\,\text{km}$). This shorter wavelength can be observed in Fig. B1, confirming that the wind speed and direction oscillations in the wind farm wake are related to an inertial oscillation.

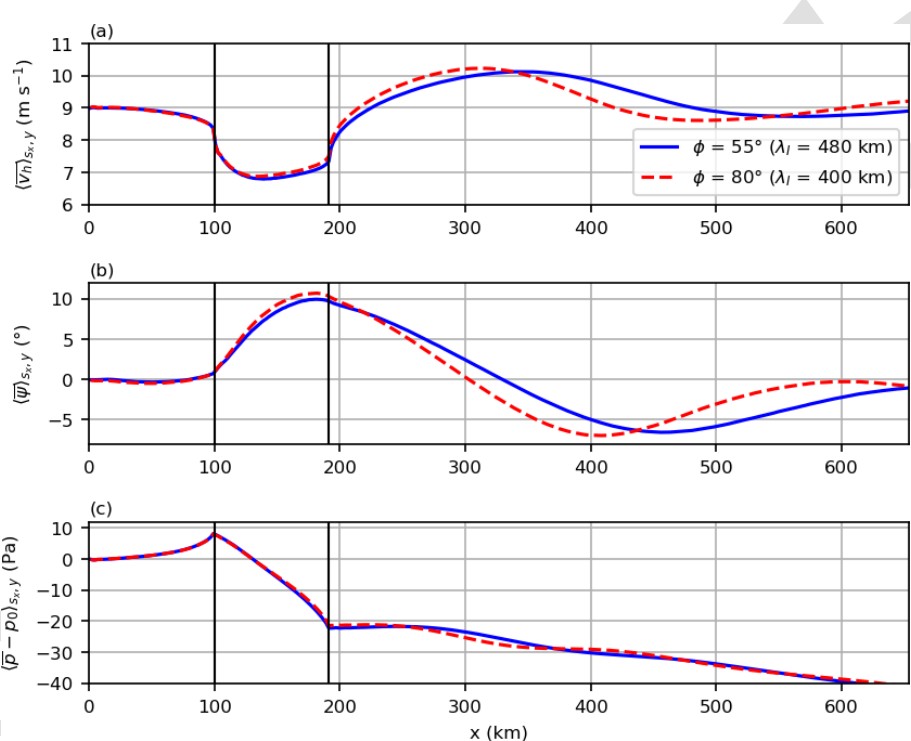

**Figure B1.** Horizontal wind speed, wind direction and relative perturbation pressure at hub height for the large wind farm case and a latitude of $\phi = 55°$ and $\phi = 80°$.

**Code and data availability.** The PALM code is available at https://gitlab.palm-model.org/releases/palm_model_system (last access: 31 March 2023). The PALM input files, additional user code and plot scripts are available at https://doi.org/10.25835/z5zxagiz (Maas, 2022). Output data are available on request.

**Competing interests.** The author has declared that there are no competing interests.

**Disclaimer.** Publisher's note: Copernicus Publications remains neutral with regard to jurisdictional claims in published maps and institutional affiliations.

**Acknowledgements.** This work was funded by the Federal Maritime and Hydrographic Agency (BSH) (grant no. 10044580) and supported by the North German Supercomputing Alliance (HLRN). Special thanks go to Siegfried Raasch for guiding the manuscript preparation. I thank Dries Allaerts and Dieter Etling for informative discussions about gravity waves and Sukanta Basu for discussions about the validity of the Boussinesq approximation. CE2

**Financial support.** This work was funded by the Federal Maritime and Hydrographic Agency (BSH) (grant no. 10044580). The publication of this article was funded by the open-access fund of Leibniz Universität Hannover.

**Review statement.** This paper was edited by Sara C. Pryor and reviewed by Dries Allaerts and two anonymous referees.

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

## Remarks from the language copy-editor

CE1    Please note that value changes require editor approval. Please provide an explanation for the editor.
CE2    Please confirm this section or let me know if any other adjustments are necessary.

## Remarks from the typesetter

TS1    In line with our house standards, one column format is necessary here.
TS2    Please provide date of last access.
TS3    Please provide date of last access.
TS4    Please provide date of last access.
TS5    Please provide date of last access and correct link.