# Peer review of "From gigawatt to multi-gigawatt wind farms: wake effects, energy budgets and inertial gravity waves investigated by large-eddy simulations"

_Wind Energy Science, 2022_

## Author Comment (AC1)

I first like to thank the reviewer for taking the time and review my manuscript. Please find my answers to the comments below. The reviewers comments are marked in black and my answers in blue.

The manuscript presents LES simulations of a small and a very large wind farm extending hundreds of kilometers. The PALM code is employed for the large scale simulations involving billions of grid points. The numerical predictions are compared and discussed in detail.

It is well written and has a potential to contribute to the state of the art in wind farm simulations and flow physics. However some predictions and the related discussions need further clarification and verification prior to publication:

- It is not stated that the x-direction is pointing south.

The x-direction is not pointing south. Since this is an idealized study, the model domain orientation is not of importance.

- The change in the horizontal mean flow direction observed in Fig 2, especially the CCW and the following CW deflections of the mean flow for the large WF needs further verification. It is also interesting that there is no visible deflection of the upstream flow in the first 60-70km range (Figs 2&3). A simulation without Coriolis force is suggested.

I think these effects are very well described in lines 205-216 and Figure 5 (crosswise tendencies):

"At the inflow all forces sum to zero and the mean flow is in a steady state. Due to the wind speed reduction upstream and inside the wind farms, the Coriolis force is reduced, so that the geostrophic pressure gradient force predominates and tends to deflect the flow counterclockwise. The vertical momentum flux divergence, however, tends to deflect the flow clockwise but this force is weaker, so that the sum of these forces is still positive. Because the wind farms are infinite in the y-direction, the perturbation pressure gradient force is parallel to the x-axis and has thus no effect on the wind direction at first. However, due to the change in wind direction further downstream inside the large wind farm, the perturbation pressure gradient force has a component perpendicular to the streamlines that tends to deflect the flow clockwise. At the end of the large wind farm the sum of all forces becomes negative, so that the flow begins to turn clockwise. Because the wind speed increases to super-geostrophic values in the wake, the Coriolis force becomes greater than the geostrophic pressure gradient force so that the flow is deflected clockwise. The most significant difference between the small and the large wind farm is that the speed deficit in the large wind farm is greater and lasts longer. This results in a greater wind direction change and thus a greater inertial wave amplitude compared to the small wind farm."

The crosswise tendencies, shown in Fig. 5, clearly prove that the cause for the deflection is the change in Coriolis force. The magnitude of the tendencies (0.0001 m/s²) also corresponds well to the change in the v-component (~1 (m/s)/(100 km)). Performing an extra simulation as an additional proof is not necessary in my opinion.

[Figure]

**Figure 5.** Crosswise forces (perpendicular to streamlines) at hub height along $x$, averaged along $y$ for the small wind farm (top) and large wind farm (bottom). Shown are the divergence of the vertical turbulent momentum flux (resolved + SGS) F, the geostrophic forcing G (difference between geostrophic pressure gradient force and Coriolis force) and the perturbation pressure gradient force P.

- In Fig 3, the reduced wind speed within the WF due to blockage and its correlation with the farm size are expected. However, the increased wind speed by about 12% 150km down in the large WF wake similarly need further verification.

It is not completely clear, what is meant by "further verification". The text (l. 187 - 200) and Fig.~4 clearly state that the speedup is related to the inertial oscillation that is triggered by the wind farm and that the amplitude is larger for the larger wind farm. Please give more specific hints about which information shall be added.

In addition to the hub height, the vertical variation of the wind speed should also be presented.

The vertical variation of wind speed and wind direction is given in Fig. 8 and is described in lines 271-295.

In addition, a simulation without the gravitational force is suggested to identify the cause and to validate the numerical implementation.

Unfortunately this comment is not clear to me. Please clarify what is meant by "without gravitational force". Does it mean without the buoyancy term, i.e. pure neutral stratified? Which cause shall be identified with this additional simulation? The speedup in the wind farm wake is related to the inertial oscillation and not to gravity.
A simulation without gravity/buoyancy is very unrealistic. Additionally, there will be no gravity waves, which is one of the main results of this paper. Thus, I do not see the necessity to do such an extra simulation.

- Similarly, in Fig.8 the BL profile at TE+120km shows that the BL flow is energized significantly up to the BL height. Such an unexpected behavior is attributed to the drop in the perturbation pressure.

This is true for the flow at the BL top. Inside the BL, however, the increase in wind speed is related to the inertial oscillation. I have added this information in line 286:
" In the far wake, one quarter of the inertial wave length ($\lambda_I /4 = 120$ km) downstream of the wind farm TEs, the wind speed in the bulk of the BL is supergeostrophic. At 300 m height the wind speed has increased to 9.2 ms−1 and 10.1 ms−1 for the small and large wind farm, respectively. **This corresponds to a wind speed increase of 0.2 and 1.1 ms−1 relative to the inflow wind speed. These values approximately correspond the amplitude of the inertial wave (see Fig.~4).**"

The velocity profile should be extended further up to see any momentum deficiency and the discussion should be extended to include what causes such a the perturbation pressure drop.

The discussion in section 3.3 (Boundary layer modification) focuses on and is limited to modifications inside the BL. A vertical extension of the profiles is thus not appropriate. The velocity-, pressure and temperature fields in the free atmosphere are determined by gravity waves, which are discussed in detail in section 3.4 (Gravity waves).

In addition, the momentum deficiency ocaused by WTs should be presented in a plot at the center plane or averaged only across the turbine (not averaged in the full y range)

Including such a plot would certainly provide more details of the flow near the wind turbines. However, the aim of this study is to focus on effects that are on the wind farm scale or larger. Thus, adding such a figure is not relevant to the outcome and conclusion and I would like to avoid it.

- The energy budget analysis is performed by integrating the quantities over the control volume of a WT. It is also not clear if the presented values in Fig 11 are averaged over all the CVs. Such an analysis gives the distribution of energy components within the CV, but not the relations between them. For a better understanding, the integrations should be performed at the control surfaces (inflow, top, outflow) of individual CVs and they should be presented as a series for a turbine row. (inflow of a downstream WT would be the outflow of the upstream WT)

The values in Figure 11 are obtained by integrating over the entire wind farm, as it is stated in the caption:
"Figure 11. Energy budgets inside control volume Ωwf , that envelops the entire small/large wind farm. [...]"
The focus of section 3.5 (Energy budget analysis) is on the energy budgets in the control volumes, i.e. the net inflow/outflow or source/sink of energy in a control volume. In- and outflow values (e.g. of the advection term A) can be much larger than their sum (budget), so that presenting them in the same figure is not possible. Thus, such an analysis would require at least two more figures. Since it is only the net energy inflow (i.e. the budget) that can be extracted by the wind turbines, I think it is reasonable to focus only on the budgets. However, the data and the scripts for the requested calculations are freely available in the cited data repository and can be used for this purpose by any interested reader.

- It is quite counter intuitive that the pressure force contributes more to the energy production than the kinetic energy of the wind as suggested in Fig 12. It even acts as a sink for downstream turbines. It definitely needs further validation and explanation.

I agree that this is a counter intuitive result. But this behavior has already been observed by Allaerts and Meyers (2017), which validates the result. I added this citation and I have now also stated more clearly that this is caused by the negative perturbation pressure gradient near the end of the farm:

"The flow acceleration at the end of the wind farms is mainly caused by the negative perturbation pressure gradient that has the highest magnitude at the wind farm TE (see Fig. 3). The energy input by the pressure gradient P thus increases towards the TE of the large wind farm and reaches 5 MW per turbine at the TE. The pressure distribution inside the wind farm is determined by wave type two of the gravity waves (see Sec. 3.4 and Fig. 9). The flow acceleration near the TE of the wind farm and the related negative net advection of KE has also been reported by Allaerts and Meyers (2017) for a 15 km long wind farm in a conventionally neutral BL."

---

## Referee Report (RR1)

The paper compares the flow behaviour and energy household of "small" and "large", infinitely wide, wind farms based on two large-eddy simulations. The size of the "large" wind farm and the corresponding large-eddy simulation is considerably larger than most LES studies of wind farms that have been performed to date. As a result of this, the author is able to show for the first time (to my knowledge) that very large wind farms trigger inertial waves in the wake of the farm. In addition to this new physical flow mechanism, the paper highlights a number of interesting differences in the energy household of "small" and "large" wind farms. The paper is also well written, the methodology is neatly described, and the results are analysed in great detail. I therefore believe the paper could be of great value to the wind energy community.

However, I also have some serious concerns about the manuscript. Most importantly, I believe the introduction fails to achieve one of it's primary purposes, namely, to put the presented research into perspective. Apart from two references to depict the size of modern wind farm projects (Herzig, 2022, and BSH, 2021), and a reference to the grand challenges paper of Veers et al. (2019) to indicate the importance of understanding wind farm flow physics, the introduction contains only one reference to relevant past work. To make matters worse, that reference is to the author's own work. This is simply unheard of. The introduction as it is now gives the impression that wind-farm flow physics and LES thereof is new and has only been explored by the author himself, while there is in fact a large volume of published studies available which this work inherently builds upon. I'm fully aware that later sections of the manuscript do include more references to relevant work, but already in the introduction the context of this work needs to be described. What other studies have looked at wind farm flow physics? What is the size of wind farms in typical LES studies? How does that compare to this work? Furthermore, the author often discusses several flow mechanisms like wake deflection, inversion layer displacement, wind-farm blockage, gravity waves, etc. (see, e.g., line 34 and line 94 for first mentioning of some of these effects), but these concepts have not been introduced properly in the manuscript. The author seems to assume that the reader is already familiar with these concepts and gives no description or proper reference to the literature. I don't think wind-farm flow physics already reached the point where it needs no introduction. In fact, some of these flow mechanisms have been discovered fairly recently and are still topic of active research. In conclusion, I believe the manuscript requires a proper introduction that describes the state-of-the-art in wind-farm LES research, puts the presented research into perspective, and explains the flow mechanisms relevant to this work.

Apart from my main comment about the introduction, I have a few more scientific and technical comments as listed below:

**Scientific comments**

1. The manuscript repeatedly talks about the size of wind farms (in terms

of the rated power) considered in the simulations. The size of the wind farm is perhaps relevant for computational resources (it shows how much computational resources are available to the author), but from a physical point of view the size is irrelevant since the farms are infinitely wide in the spanwise direction. For the flow physics of infinitely wide wind farms, what matters is the streamwise length of the farm or the number of turbine rows. In this respect, I have the following specific comments:

(a) Lines 7-8, 42-43, 77-78: The rated power of infinitely wide wind farms is meaningless as it is physically irrelevant and depends on the choice of turbine spacing and spanwise extent of the numerical domain, as these two parameters determine the number of turbine columns resolved within the simulation. Please use a more relevant parameter to distinguish the different cases.

(b) Line 79: Why is the large wind farm simulation (16 turbine rows) twice as wide as the small one (8 turbine rows) if you are using periodic BC anyway? What is the reasoning behind this?

(c) Line 92: "Note that the small wind farm is already as big as the largest wind farms of most other LES studies, e.g. Wu and Porté-Agel (2017) (length 19.6 km, rated power 0.36 GW) ..." Comparison of power for infinitely wide wind farms is meaningless because it depends on how many columns are resolved (see also previous comment). Your study has 8 and 16, Allaerts&Meyers had 9, Wu and Porté-Agel had 5...

(d) Line 48-49: "To my knowledge this is the second largest wind farm LES study in terms of domain size and wind farm power after the study of Maas and Raasch (2022)." Why is it relevant that this is the second largest wind farm LES study? Instead of just mentioning the ranking, a quantitative comparison with wind farm size in other LES studies would be more interesting.

2. The inertial wave developing in the wake of the farm is quite an interesting finding, and it leads to some strong statements related to wind speed ups and impact on downstream located wind farms (see, e.g., line 12-13 and line 173-174). If these statements hold in general, they could have strong implications for wind energy deployment. Therefore, I'm surprised to see that the study is based on only two simulations, and I share past reviewers concern about verification/validation of results. I do agree with the author that it is not so easy to turn off gravity or the Coriolis force, but there are other options to increase confidence in the presented results. For example, I assume that the amplitude of the inertial wave depends on how much the flow decelerates inside the farm, so you should be able to see different wave amplitudes with different wind farm lengths or even different wind farm layouts. Maybe it is worthwhile to consider a wind farm of intermediate length or a different layout. Alternatively, if the flow behaviour you are seeing is indeed an inertial oscillation triggered by the wind farm, the

wave length should be independent of wind farm length and depend on the Coriolis parameter. You could consider a different latitude (boundary-layer height is governed by the temperature structure and subsidence so no issue there) and see whether the wave length changes accordingly.

3. I have several questions about the wavelength analysis from line 325 onward. First of all, I believe equation 14 is incorrect. Based on equation 12 and 13 and assuming that you define the "absolute wavelength $\lambda$" as the wave length in the direction of phase propagation, i.e. $\lambda = \lambda_x \cos \alpha$ (note that this is nowhere defined!), I find that the absolute wavelength should be given by

$$\lambda = \frac{1}{\sqrt{1 + \frac{f^2 \sin^2 \alpha}{N^2 \cos^2 \alpha}}} \frac{2\pi U}{N} \tag{1}$$

Second, looking at the first two entries in table 1, how is it possible that $\lambda$ for wave type 1 and 2 (small wind farm) is the same for two different inclination angles $\alpha$? According to eq. 14, there should be a unique relation between $\alpha$ and $\lambda$ for U, N and f constant.

Thirdly, eq. 14 has two unknowns: the inclination angle and the wave length. The manuscript does not mention how you come to the results in table 1? Did you measure the wavelength in the simulation and then calculated the inclination angle? Or did you approach it the other way around, estimating the inclination angle from the figures and then calculating the wave length based on eq. 14?

**Minor/technical comments**

1. Eq.3: In the last term on the right-hand side, the subscript of $u$ should be $j$ instead of $i$.

2. Line 105: "... which is enough for resolving the gravity waves with a wave length of approximately 5 km." How did you calculate that wave length? This is coming out of nowhere. Please explain, or refer to later section.

3. Can you briefly describe the radiation boundary conditions of Miller and Thorpe (1981) and Orlanski (1976) in the paper? As the inertial oscillation triggered by the wind farms is a wave phenomenon itself, with a very large wave length (see figure 2 and 3), how do you know that the flow results are not affected by the outflow boundary? Clearly, in figure 3a, the wave extends all the way down to the outflow boundary. If you would put that boundary closer or further away, do you still obtain the same wave properties?

4. Line 119: What values are used to come to the advection distance of the convective time scale? Also, please explicitly mention the definition of the convective velocity scale $w^*$ and the values used to calculate the quantity.

5. Section 2.3: Please explain the particular choice of surface heating and boundary-layer height (for instance, a boundary-layer height of 600m in convective conditions seems quite low).

6. Line 130-132: "The initial horizontal velocity is set to the geostrophic wind (Ug,Vg) = (9.011,-1.527) ms-1, resulting in a steady-state hub height wind speed of 9.0 ms-1 that is aligned with the x-axis." How did you find this particular geostrophic wind speed and direction? Did you find this by trial and error or did you use a particular method (e.g., a wind speed/angle controller)?

7. Line 144: "... so that the inertial oscillation has decayed ..." I assume you are referring to an inertial oscillation in time that occurs after the simulation is initialized? Please explain why an inertial oscillation is triggered. Please also clearly mention that you are talking about an inertial oscillation in time to avoid confusion with the inertial oscillation observed downstream of large wind farms.

8. Line 182: "Further downstream the wind turn clockwise ..." → "Further downstream, the wind turns clockwise ..."

9. Line 209: "Because the wind farms are infinite in the y-direction, the perturbation pressure gradient force is parallel to the x-axis and has thus no effect on the wind direction at first." Is this because the perturbation pressure is due to gravity waves, which are uniform in y direction because the wind farm is infinitely wide? This statement needs more explanation.

10. Line 220: "If the Rossby number [based on the wind farm length] is smaller or close to 1, Coriolis effects become dominant ...". It is not necessarily true that Coriorlis forces become dominant at Rossby number close to 1. I agree that Coriolis forces become dominant as the Rossby number decreases, but you cannot claim (at least not based on two simulations) that the tipping point is for Rossby equal to 1.

11. Figure 6: It is surprising to see that neither TKE nor TI shows an oscillation. I would expect at least one of the two to be affected by the oscillatory behaviour in wind speed, as TI is related to TKE normalized by that same wind speed. How do you explain this? Further, on lines 235-236 you say that "TKE is greater in the small wind farm, because the wind speed is greater." How is TKE related to the wind speed magnitude? TKE production is related to wind speed gradients, so I don't see how TKE is directly related to wind speed magnitude.

12. In lines 248-250 you say that "TI does not show the oscillatory behavior that the wind speed and direction show because turbulence has time scales that are orders of magnitude smaller than that of the mean flow and therefore hardly affected by the Coriolis force." Are you saying that the turbulence time scales are orders of magnitude smaller than the inertial oscillation period, and that therefore TKE rapidly adapts to changes

in wind speed such that TI (or TKE) is constant? Please clarify this statement.

13. Line 252: "The last two sections ..." $\rightarrow$ "The previous two sections ..."

14. Line 349: "...which approximately corresponds the the length of the small wind farm ..." should be "...which approximately corresponds *to* the length of the small wind farm ..."

15. Line 356: "Due to the large ratio of $L_{wf}/L_s$ in the large wind farm case, wave type one and two can be clearly distinguished in this study." Why would a larger ratio of wind farm length to Scorer parameter make the wave more distinguishable?

16. Section 3.5.1: How did you scale the budget terms from $W\rho^{-1}$ to MW per turbine? Did you assume $\rho = 1.17\,\mathrm{kg\,m^{-3}}$ like in eq. 11?

17. Caption of figure A1: apprxomation $\rightarrow$ approximation

---

## Author Response (AR2)

In this second response to the comments of referee #1 I provide more detailed answers to some of the comments, as it was requested by the associate editor. The reviewers comments are marked in black and my answers in blue. The additional, more detailed answers are marked in green.

- The change in the horizontal mean flow direction observed in Fig 2, especially the CCW and the following CW deflections of the mean flow for the large WF needs further verification.

I think these effects are very well described in lines 205-216 and Figure 5 (crosswise tendencies):

"At the inflow all forces sum to zero and the mean flow is in a steady state. Due to the wind speed reduction upstream and inside the wind farms, the Coriolis force is reduced, so that the geostrophic pressure gradient force predominates and tends to deflect the flow counterclockwise. The vertical momentum flux divergence, however, tends to deflect the flow clockwise but this force is weaker, so that the sum of these forces is still positive. Because the wind farms are infinite in the y-direction, the perturbation pressure gradient force is parallel to the x-axis and has thus no effect on the wind direction at first. However, due to the change in wind direction further downstream inside the large wind farm, the perturbation pressure gradient force has a component perpendicular to the streamlines that tends to deflect the flow clockwise. At the end of the large wind farm the sum of all forces becomes negative, so that the flow begins to turn clockwise. Because the wind speed increases to super-geostrophic values in the wake, the Coriolis force becomes greater than the geostrophic pressure gradient force so that the flow is deflected clockwise. The most significant difference between the small and the large wind farm is that the speed deficit in the large wind farm is greater and lasts longer. This results in a greater wind direction change and thus a greater inertial wave amplitude compared to the small wind farm."

It is also interesting that there is no visible deflection of the upstream flow in the first 60-70km range (Figs 2&3).

The deflection inside the wind farm and the wake is caused by the reduction in wind speed which results in a reduction of the Coriolis force, which is linearly proportional to the wind speed. Since there is no wind speed deficit in the first 60 - 70 km of the domain, there is also no deflection.

A simulation without Coriolis force is suggested.

The crosswise tendencies, shown in Fig. 5, clearly prove that the cause for the deflection is the change in Coriolis force. The magnitude of the tendencies ($0.0001$ m/s$^2$) also corresponds well to the change in the v-component (~1 (m/s)/(100 km)). Performing an extra simulation as an additional proof is not necessary in my opinion.

[Figure]

**Figure 5.** Crosswise forces (perpendicular to streamlines) at hub height along $x$, averaged along $y$ for the small wind farm (top) and large wind farm (bottom). Shown are the divergence of the vertical turbulent momentum flux (resolved + SGS) F, the geostrophic forcing G (difference between geostrophic pressure gradient force and Coriolis force) and the perturbation pressure gradient force P.

I would like to add here, that I am of course happy to perform an additional simulation without Coriolis force to underpin the results, if the reviewer insists. However, setting up a simulation without Coriolis force raises some problems and questions that I would like to name and explain shortly here:

- The forcing of the flow, that is otherwise done with the geostrophic wind, has to be done with a streamwise pressure gradient that opposes the friction in the boundary layer. This is possible in PALM but raises some other problems and questions:
- Since there is no friction in the free atmosphere, the pressure gradient has to be zero there, otherwise the flow will not reach a steady state. This raises the question of the right choice of the vertical profile of the pressure gradient, which can be arbitrarily chosen.
- Depending on the above named choice of the pressure gradient profile, the boundary layer height and also other profiles such as wind speed, wind direction and momentum flux profiles may differ significantly from the original simulation (with Coriolis force). E.g. the wind direction will be constant with height (no veer).
- As shown above, many other parameters than only the Coriolis force would be different between the original and the additional simulation. Consequently, making direct comparisons between these simulations is problematic and derived conclusions are of limited validity, in my opinion. E.g. it can not be distinguished whether the wake deflection is caused by wind veer or by the Coriolis force, because **both** are not present in the additional simulation.

- In Fig 3, the reduced wind speed within the WF due to blockage and its correlation with the farm size are expected. However, the increased wind speed by about 12% 150km down in the large WF wake similarly need further verification.

It is not completely clear, what is meant by "further verification". The text (l. 187 - 200) and Fig.~4 clearly state that the speedup is related to the inertial oscillation that is triggered by the wind farm and that the amplitude is larger for the larger wind farm. Please give more specific hints about which information shall be added.

    In addition to the hub height, the vertical variation of the wind speed should also be presented.

The vertical variation of wind speed and wind direction is given in Fig. 8 and is described in lines 271-295.

    In addition, a simulation without the gravitational force is suggested to identify the cause and to validate the numerical implementation.

Unfortunately this comment is not clear to me. Please clarify what is meant by "without gravitational force". Does it mean without the buoyancy term, i.e. pure neutral stratified? Which cause shall be identified with this additional simulation? The speedup in the wind farm wake is related to the inertial oscillation and not to gravity.
A simulation without gravity/buoyancy is very unrealistic. Additionally, there will be no gravity waves, which is one of the main results of this paper. Thus, I do not see the necessity to do such an extra simulation.

I would like to add some more thoughts on the requested additional simulation without gravity, to explain my last comment in more detail:
Here, I assume that "without gravity" means a simulation without buoyancy forces and thus a pure neutral stratification up to the domain top. First, I agree that performing such a simulation would give some more interesting insights, e.g. it will prove whether the pressure distribution and the blockage effect is caused by the gravity waves or also by other effects.
However, a comparison between this additional simulation and the original simulation is again problematic, because many parameters change at the same time, e.g.:
- The original simulation consists of a convective boundary layer. Without buoyancy forces in the additional simulation the boundary layer will be neutrally stratified. This results in a different boundary layer height and in different profiles of the wind speed, wind direction and turbulent fluxes.
- The above named changes will affect the wake flow and the blockage effect.
- Since a stable stratification and the buoyancy force as restoring force is needed to support the formation of gravity waves, no gravity waves will form in the - now neutrally stratified -

free atmosphere. This will presumably have a significant effect on the pressure and wind speed distribution at hub height.

- In reality the free atmosphere is always stably stratified. I therefore used a stably stratified free atmosphere with a lapse rate corresponding to the international standard atmosphere to obtain a simulation that is as realistic as possible.

- Similarly, in Fig.8 the BL profile at TE+120km shows that the BL flow is energized significantly up to the BL height. Such an unexpected behavior is attributed to the drop in the perturbation pressure.

This is true for the flow at the BL top. Inside the BL, however, the increase in wind speed is related to the inertial oscillation. I have added this information in line 286:
" In the far wake, one quarter of the inertial wave length ($\lambda I /4 = 120$ km) downstream of the wind farm TEs, the wind speed in the bulk of the BL is supergeostrophic. At 300 m height the wind speed has increased to 9.2 ms−1 and 10.1 ms−1 for the small and large wind farm, respectively. **This corresponds to a wind speed increase of 0.2 and 1.1 ms−1 relative to the inflow wind speed. These values approximately correspond the amplitude of the inertial wave (see Fig.~4).**"

The velocity profile should be extended further up to see any momentum deficiency and the discussion should be extended to include what causes such a the perturbation pressure drop.

The discussion in section 3.3 (Boundary layer modification) focuses on and is limited to modifications inside the BL. A vertical extension of the profiles is thus not appropriate. The velocity-, pressure and temperature fields in the free atmosphere are determined by gravity waves, which are discussed in detail in section 3.4 (Gravity waves).

Please find below the vertical profiles that extend up to the domain top. As can be seen, the profiles have a sinusoidal shape and are thus related to the gravity waves in the free atmosphere. The amplitudes decay above 5000 m due to the Rayleigh-damping layer. If you insist that these profiles shall be included in the article, then I would suggest to put them into section 3.4 (gravity waves).

[Figure]

In addition, the momentum deficiency ocaused by WTs should be presented in a plot at the center plane or averaged only across the turbine (not averaged in the full y range)

Including such a plot would certainly provide more details of the flow near the wind turbines. However, the aim of this study is to focus on effects that are on the wind farm scale or larger. Thus, adding such a figure is not relevant to the outcome and conclusion and I would like to avoid it.

- The energy budget analysis is performed by integrating the quantities over the control volume of a WT. It is also not clear if the presented values in Fig 11 are averaged over all the CVs. Such an analysis gives the distribution of energy components within the CV, but not the relations between them. For a better understanding, the integrations should be performed at the control surfaces (inflow, top, outflow) of individual CVs and they should be presented as a series for a turbine row. (inflow of a downstream WT would be the outflow of the upstream WT)

The values in Figure 11 are obtained by integrating over the entire wind farm, as it is stated in the caption:
"Figure 11. Energy budgets inside control volume Ωwf , that envelops the entire small/large wind farm. [...]"
The focus of section 3.5 (Energy budget analysis) is on the energy budgets in the control volumes, i.e. the net inflow/outflow or source/sink of energy in a control volume. In- and outflow values (e.g. of the advection term A) can be much larger than their sum (budget), so that presenting them in the same figure is not possible. Thus, such an analysis would require at least two more figures. Since it is only the net energy inflow (i.e. the budget) that can be extracted by the wind turbines, I think it is reasonable to focus only on the budgets. However, the data and the scripts for the requested calculations are freely available in the cited data repository and can be used for this purpose by any interested reader.

Please find below the requested plots of the input-/ output terms of each control volume.

[Figure]

The first figure shows the energy input by advection of kinetic energy through the left and right control volume boundaries. Note that $A_{x,right}$ has a negative sign (in the legend) and is thus a sink. As can be seen, the absolute values are much larger then the differences (the budget). This is mainly

caused by the choice of the control volumes: The turbine only occupies a relatively small portion of the crosswise extend of the control volume and thus most of the flow that leaves the control volume on the right side is not part of the decelerated wake flow.

The second figure shows the vertical advection as well as the horizontal and vertical fluxes. These terms are at least one magnitude smaller and thus require an extra figure. The advection and fluxes through the bottom boundary are much smaller then the respective top values, because they are limited due to the proximity to the surface (bottom boundary is at z = 60 m = 3rd grid point). The evolution of the horizontal turbulent fluxes $F_x$ is similar to the evolution of the TKE (see Figure 6 of the manuscript). Since the wind turbines generate additional TKE, more TKE leaves the CV thorugh the right boundary than it enters from the left boundary. After approx. 5 rows the TKE has reached a steady value and the net in/outflow is zero.

Please let me know if you think that these figures are worth to be included in the manuscript.

- It is quite counter intuitive that the pressure force contributes more to the energy production than the kinetic energy of the wind as suggested in Fig 12. It even acts as a sink for downstream turbines. It definitely needs further validation and explanation.

I agree that this is a counter intuitive result. But this behavior has already been observed by Allaerts and Meyers (2017), which validates the result. I added this citation and I have now also stated more clearly that this is caused by the negative perturbation pressure gradient near the end of the farm:

"The flow acceleration at the end of the wind farms is mainly caused by the negative perturbation pressure gradient that has the highest magnitude at the wind farm TE (see Fig. 3). The energy input by the pressure gradient P thus increases towards the TE of the large wind farm and reaches 5 MW per turbine at the TE. The pressure distribution inside the wind farm is determined by wave type two of the gravity waves (see Sec. 3.4 and Fig. 9). The flow acceleration near the TE of the wind farm and the related negative net advection of KE has also been reported by Allaerts and Meyers (2017) for a 15 km long wind farm in a conventionally neutral BL."

---

## Author Response (AR3)

I thank referee 1 (anonymous) for his or her additional comments and I regret that my last answers were not satisfactory. Below I provide detailed answers to the reviewers comments and I additionally provide equations or sketches to avoid any misunderstandings. I performed two additional large wind farm simulations, one with a different latitude and one with a different domain length to verify the results. I hope that the new results and the additional explanations are satisfactory to the reviewer.

The reviewers comments are marked in black and my answers in blue.

The responses given to my comments are far from being satisfactory
and some did not make sense:

> The x-direction is not pointing south. Since this is an idealized
> study, the model domain orientation is not of importance.

The Coriolis force is latitude and direction dependent.
Such a response does not make sense.

The reviewer is right, the Coriolis force depends on the latitude and the orientation of the model domain. However, the model domain orientation does not affect the simulations results, as I want to show with the model equations. The effect of different model domain rotations on the flow tendencies can be seen in the equations (1) to (13) on the following page.

Equation (2) to (4) show that a rotation of the model domain only affects f1 and f2 (the two horizontal parts of the Coriolis parameter vector) and not f3 (the vertical part of the vector).

In equation (7) the vertical velocity (u3) affects the u1-velocity via f2. In equation (10) the vertical velocity (u3) affects the u2-velocity via f1. Thus, if the model domain orientation changes, then only the effect of the vertical velocity component (u3) may affect the two horizontal velocity components (u1 and u2). Since the vertical velocity is approximately zero in the far wake at hub height, u1 and u2 will not be affected by a model rotation. Consequently, the inertial oscillation will also be unaffected by a model domain rotation.

In equation (13) it can be seen, that both horizontal velocity components (u1 and u2) have an effect on the tendency for the vertical velocity via f2 and f1. However, since this tendency will be equal over the entire domain width (Ly) and because the flow is constraint by the proximity of the surface, no vertical motions will occur (the Coriolis induced tendencies will be opposed by a vertical perturbation pressure gradient).

That the properties of inertial oscillations do not depend on the mean wind direction (and thus f1 and f2) can also be deduced from the fact that text books or publications that deal with that topic only consider the vertical part of the Coriolis parameter (f3) and do not consider f1 or f2 (e.g. Stull, 2009, p. 524 and Baas et al., 2012, as cited in the manuscript).

The Coriolis term in PALM's momentum conservation equation is:

$$\frac{\partial u_i}{\partial t} = \dots - \epsilon_{ijk} f_j u_k + \epsilon_{i3j} f_3 u_{g,j} \dots , \tag{1}$$

with the Coriolis parameter vector:

$$f_i = (f_1, f_2, f_3) = (2\Omega \cos\phi \sin\gamma, 2\Omega \cos\phi \cos\gamma, 2\Omega \sin\phi) , \tag{2}$$

where $\Omega$ is the rotational speed of the Earth, $\phi$ is the latitude and $\gamma$ is the clockwise rotation angle of the model domain ($\gamma = 0$, if $x$ pointing to the East). In the presented simulations $\gamma = 0$, so that:

$$f_i = (0, 2\Omega \cos\phi, 2\Omega \sin\phi) . \tag{3}$$

For a model domain rotation of e.g. $\gamma = 90°$, $f_i$ is:

$$f_i = (2\Omega \cos\phi, 0, 2\Omega \sin\phi) . \tag{4}$$

Thus, a rotation of the model domain affects only $f_1$ and $f_2$.

Tendencies in the x-direction are then given by:

$$\frac{\partial u_1}{\partial t} = -\epsilon_{1jk} f_j u_k + \epsilon_{13j} f_3 u_{g,j} \tag{5}$$

$$\frac{\partial u_1}{\partial t} = -\epsilon_{123} f_2 u_3 - \epsilon_{132} f_3 u_2 + \epsilon_{132} f_3 u_{g,2} \tag{6}$$

$$\frac{\partial u_1}{\partial t} = -f_2 u_3 + f_3(u_2 - u_{g,2}) \tag{7}$$

Tendencies in the y-direction are given by:

$$\frac{\partial u_2}{\partial t} = -\epsilon_{2jk} f_j u_k + \epsilon_{23j} f_3 u_{g,j} \tag{8}$$

$$\frac{\partial u_2}{\partial t} = -\epsilon_{213} f_1 u_3 - \epsilon_{231} f_3 u_1 + \epsilon_{231} f_3 u_{g,1} , \tag{9}$$

$$\frac{\partial u_2}{\partial t} = f_1 u_3 + f_3(u_1 - u_{g,1}) \tag{10}$$

Tendencies in the z-direction are given by:

$$\frac{\partial u_3}{\partial t} = -\epsilon_{3jk} f_j u_k + \epsilon_{33j} f_3 u_{g,j} \tag{11}$$

$$\frac{\partial u_3}{\partial t} = -\epsilon_{312} f_1 u_2 - \epsilon_{321} f_2 u_1 + \epsilon_{231} f_3 u_{g,1} , \tag{12}$$

$$\frac{\partial u_3}{\partial t} = -f_1 u_2 + f_2 u_1 \tag{13}$$

> • The change in the horizontal mean flow direction observed in
> Fig 2, especially the CCW and the following CW deflections of the
> mean flow for the large WF needs further verification.

> I think these effects are very well described in lines 205-216 and
> Figure 5 (crosswise tendencies):

The verification needs further analysis and additional data.
Further explanation of the results which are already in question
does not verify itself.

I performed two additional simulations with the large wind farm to verify this result:

1. A simulation with an even longer domain length (655.36 km instead of 409.6 km)

The results of this simulation show that after the CCW and CW rotation another CCW and CW rotation
follows, proving that this is an oscillation. Furthermore, the additional simulation shows that the outflow
boundary does not affect the results.

[Figure]

2. A simulation with a different latitude (80° N), resulting in a reduction of the wavelength.

The wavelength of the inertial oscillation is given by the oscillation period and the geostrophic wind speed
(please refer to the next page for the equations). For the larger latitude, the oscillation period is shorter and thus
the wavelength is also shorter.

Oscillation period of the inertial oscillation:

$$T = \frac{2\pi}{f_3} = \frac{2\pi}{2\Omega \sin\phi} \tag{16}$$

$$\begin{aligned} T(\phi = 55°) &= 14.6 \text{ h} \\ T(\phi = 80°) &= 12.1 \text{ h} \end{aligned} \tag{17}$$

Wavelength of inertial wave:

$$\begin{aligned} \lambda_I &= GT \\ \lambda_I(\phi = 55°) &= 480 \text{ km} \\ \lambda_I(\phi = 80°) &= 400 \text{ km} \end{aligned} \tag{18}$$

with the magnitude of the geostrophic wind:

$$G = \sqrt{u_{g,1}^2 + u_{g,2}^2} = 9.1 \text{ ms}^{-1} \tag{19}$$

These wavelengths correspond very well to the wavelengths that can be measured in the original simulation (phi=55°) and the additional simulation (phi=80°), e.g. $\lambda/2 = 200$ km for lat = 80°:

[Figure]

This result confirms, that the change in wind speed and direction in the wind farm wake is related to an inertial oscillation. I have created an additional appendix B that contains the last Figure and a short description and explanation of the comparison. Please have a look at the diff document.

> • The change in the horizontal mean flow direction observed in
> It is also interesting that there is no visible deflection of the
> upstream flow in the first 60-70km range (Figs 2&3).

> The deflection inside the wind farm and the wake is caused by the
> reduction in wind speed which results in a reduction of the Coriolis
> force, which is linearly proportional to the wind speed. Since there
> is no wind speed deficit in the first 60 - 70 km of the domain,
> there is also no deflection.

Such a response again does not make sense. As stated the Coriolis force is a function of velocity, not velocity/speed deficit. If there is a flow under the Coriolis force there should be a continuous deflection. The need for a simulation without the Coriolis effect now becomes even more necessary.

It is correct that the Coriolis force is a function of the velocity and not of the velocity deficit. However, in the PALM model the effects of the Coriolis force and the synoptic-scale pressure gradient force are modeled by the formulation of the geostrophic wind (geostrophic forcing). The geostrophic wind velocity is the wind velocity at which Coriolis force and synoptic-scale pressure (ps) gradient are in balance:

$$u_{g,1} = -\frac{1}{\rho f_3}\frac{\partial p_s}{\partial y} \tag{14}$$

$$u_{g,2} = +\frac{1}{\rho f_3}\frac{\partial p_s}{\partial x} \tag{15}$$

A Coriolis force induced deflection (or tendency) in the x- or y-direction thus only occurs, of the flow velocity deviates from the geostrophic wind, i.e. if the term $(u_2 - u_{g,2})$ in equation (7) or the term $(u_1 - u_{g,1})$ in equation (10) is non-zero ("geostrophic departure", Stull, 2009, p. 524).

At the inflow, inside the BL, there is a small deviation from the geostrophic wind, causing a small accelerating tendency. However, this tendency is exactly compensated by the frictional force (vertical momentum flux divergence), as sketched on the following page. Thus all forces sum to zero and the flow is in a steady state (which is achieved by the long simulation time of the precursor simulation of which the profiles are used for the inflow boundary of the main simulation).

A flow deflection (or acceleration in any direction) only occurs if the forces are not in balance. Such a case is also sketched on the following page. In this example the wind velocity is larger than in the steady-state case, resulting in a larger Coriolis force and thus there is a net force to the south leading to a clockwise deflection.

| Case | Velocities and forces |
|------|----------------------|

[Figure]

**Mean flow is in steady state: no resulting force and no deflection.**

Velocities:

$\vec{u}$

$\vec{u_g}$

$(u_2 - u_{g,2})$

Forces in reality:

Forces in PALM:

Pressure gradient

Friction

Coriolis

Forces sum to zero if flow is in steady state

Geostrophic forcing = pressure gradient + Coriolis force = $f_3(u_2 - u_{g,2})$

Friction

**Mean flow is not in steady state: higher wind speed, southward deflection.**

Velocities:

$\vec{u}$

$\vec{u_g}$

$(u_2 - u_{g,2})$

$(u_1 - u_{g,1})$
+ 25 %

Forces in reality:

Forces in PALM:

Pressure gradient

Friction

Resulting force

Coriolis

Forces sum not to zero, deflection to the South

+ 25 %

$f_3(u_2 - u_{g,2})$

Friction

Resulting force

$f_3(u_1 - u_{g,1})$

Geostrophic forcing

> It is not completely clear, what is meant by "further
> verification". The text (l. 187 - 200) and Fig.~4 clearly state that
> the speedup is related to the inertial oscillation that is triggered
> by the wind farm and that the amplitude is larger for the larger wind
> farm. Please give more specific hints about which information shall
> be added.

Again, a verification needs further analysis and additional new data.

As mentioned further above, I have performed an additional simulation with a different latitude and thus a different wavelength which verifies that the oscillation is an inertial oscillation.

Inertial oscillations are, in general due to the interaction between
the Coriolis force and gravity. If there is no gravity, there would
be no inertial oscillations.

I kindly ask the reviewer to check this statement because I think that it is wrong. For example, in the chapter 12.5.3 "Inertial Oscillation" in the textbook "Boundary Layer Meteorology" (Stull, 2009),  gravity is not mentioned in any sentence and gravity is also not present in any of the equations.

The extended vertical velocity profiles provided in the response and
a brief discussion on the vertical momentum transfer should be
included in the manuscript.

I have added this figure as figure 10 in the manuscript and also added a short discussion about it. Please refer to the diff document.

On the other hand, the inertial/gravity
waves would be best presented with 2D streamlines on a vertical plane.
They should also be included for the reader to better understand the
flow field developing over the WFs.

I agree with the reviewer that streamlines would show the gravity waves in a more intuitive way. However, the vertical displacement amplitude is only approximately 100 m, which is only 1 % of the vertical scale of the figure 9 (10 km). Thus, the waves will not be visible in the streamlines but the streamlines will rather be horizontal lines. Consequently, showing streamlines would require an extra figure with an even more exaggerated vertical scale. I do not think that this a good solution, because the manuscript already has a lot of figures (18) and because I think that it does not add valuable information for the results of the paper.

The figure which "shows the energy input by advection of kinetic
energy through the left and right control volume boundaries", that
is, the kinetic energy flux, should only include just one "line",
since the left boundary for one turbine is the right boundary for
the upstream turbine. (By the way, a negative flux is not a sink,
but is an influx, as the positive flux is an outflux by convention.
The legend is in error.)

This is true, the energy flux on the right boundary of a control volume is the energy flux through the left boundary of the next control volume. It is also true that the sign of the flux must be changed, because it points in the same direction.

In the figure, the initial trend is expected, however, the increased kinetic energy flux after about the

20th turbine row needs explanation. It suggests that the turbines
are not extracting energy but acting line an energy source!

The fact that there is an increase in the advection of KE (flux divergence) does not mean that the turbines are energy sources. As shown in Fig. 11, there are other energy sources that compensate the divergence of the KE advection. These sources are volume sources (energy is produced inside the control volume) and surface sources (energy flows into the control volume via the boundaries).

The volume sources (or sinks) are (as indicated by the volume integral in eq. 16 of the manuscript):
- P: Energy input by mean perturbation pressure gradients
- G: Energy input by geostrophic forcing
- B: Energy input by buoyancy forces
- D: Dissipation
- W: Energy extraction by wind turbines

Surface sources (or sinks) are (as indicated by the surface integrals in eq. 19 and 20 of the manuscript):
- A: advection of KE
- F: turbulent fluxes

The increase in A towards the end of the wind farm is related to a flow acceleration, which is mainly caused by a negative perturbation pressure gradient (term "P"). This is also explained in the manuscript (line 446):

"The flow acceleration at the end of the wind farms is mainly caused by the negative perturbation pressure gradient that has the highest magnitude at the wind farm TE (see Fig. 3)."

It is not clear what the second figure exactly shows.

The second Figure shows the vertical advection, the horizontal and vertical turbulent fluxes through the control volume boundaries, as requested by the reviewer: "For a better understanding, the integrations should be performed at the control surfaces (inflow, top, outflow) of individual CVs... "

If the horizontal and vertical fluxes refer to the mass fluxes, they are
not much relevant since the conservation of mass should be satisfied.

The horizontal and vertical fluxes refer to turbulent energy fluxes as defined in the manuscript. The transport of KE by the mass flux is part of the term A (advection of KE). Anyhow, conservation of energy should be satisfied, so all energy sources and sinks should sum to zero. Reasoning that the terms are not relevant, because energy conservation should be satisfied anyhow, would make the entire energy budget analysis obsolete.

But the kinetic energy fluxes from the top and the bottom boundaries
being insignificant, it makes the previous question more significant:
Where do the downstream turbines gain kinetic energy from? Pressure?

Yes, from pressure and from other energy sources (geostrophic forcing, vertical fluxes ...), see volume sources and surface forces described above. These energy sources are discussed in lines 415 - 431 in the manuscript.

> I agree that this is a counter intuitive result. But this behavior has
> already been observed by Allaerts and Meyers (2017), which validates
> the result. I added this citation and I have now also stated more
> clearly that this is caused by the negative perturbation pressure
> gradient near the end of the farm:

Pressure/perturbation pressure is a variable in the equations solved. Blaming a solution variable for an unexpected result is another way of saying "this is what I got out of the code". The question is why the solution produces such high presure gradient at the downstream boundary. Could there be something wrong in the implementation of the boundary conditions? Could there be a way to test it out?

I do not blame the perturbation pressure for an unexpected result.

The perturbation pressure distribution in the wind farm is related to the gravity waves in the overlying free atmosphere, as can be clearly seen in Fig. 9 g and h. Special care has been taken in this study to corectly capture these gravity waves by choosing a large domain height and appropriate Rayleigh damping parameters.

[Figure]

**Figure 9.** Vertical cross sections of the horizontal wind speed (a,b), wind direction (c,d), vertical wind speed (e,f), perturbation pressure (g,h) and potential temperature (i,j). The quantities are averaged in time and along $y$ and the respective mean inflow is subtracted. Vertical lines

The outflow boundary is very far away from the wind farm trailing edge, further away then in any other wind farm LES study, to my knowledge. Since this is no proof that the BCs at the outflow do not affect the pressure distribution in the wind farm, I performed an additional simulation with a 250 km longer domain (see figure below). As can be seen, the effect on the pressure distribution in the wind farm is insignificant.

I also would like to add that a negative perturbation pressure gradient and the related energy input has also been observed and described by other studies, as cited in the manuscript, e.g. Allaerts and Meyers (2017) in line 487.

[Figure]

I thank referee 2 (Dries Allaerts) very much for thoroughly checking the manuscript and providing very constructive comments and important corrections. Please find my answers to the comments below. The reviewers comments are marked in black and my answers in blue.

The paper compares the flow behaviour and energy household of "small" and "large", infinitely wide, wind farms based on two large-eddy simulations. The size of the "large" wind farm and the corresponding large-eddy simulation is considerably larger than most LES studies of wind farms that have been performed to date. As a result of this, the author is able to show for the first time (to my knowledge) that very large wind farms trigger inertial waves in the wake of the farm. In addition to this new physical flow mechanism, the paper highlights a number of interesting differences in the energy household of "small" and "large" wind farms. The paper is also well written, the methodology is neatly described, and the results are analysed in great detail. I therefore believe the paper could be of great value to the wind energy community.

However, I also have some serious concerns about the manuscript. Most importantly, I believe the introduction fails to achieve one of it's primary purposes, namely, to put the presented research into perspective. Apart from two references to depict the size of modern wind farm projects (Herzig, 2022, and BSH, 2021), and a reference to the grand challenges paper of Veers et al. (2019) to indicate the importance of understanding wind farm flow physics, the introduction contains only one reference to relevant past work. To make matters worse, that reference is to the author's own work. This is simply unheard of. The introduction as it is now gives the impression that wind-farm flow physics and LES thereof is new and has only been explored by the author himself, while there is in fact a large volume of published studies available which this work inherently builds upon. I'm fully aware that later sections of the manuscript do include more references to relevant work, but already in the introduction the context of this work needs to be described. What other studies have looked at wind farm flow physics? What is the size of wind farms in typical LES studies? How does that compare to this work? Furthermore, the author often discusses several flow mechanisms like wake deflection, inversion layer displacement, wind-farm blockage, gravity waves, etc. (see, e.g., line 34 and line 94 for first mentioning of some of these effects), but these concepts have not been introduced properly in the manuscript. The author seems to assume that the reader is already familiar with these concepts and gives no description or proper reference to the literature. I don't think wind-farm flow physics already reached the point where it needs no introduction. In fact, some of these flow mechanisms have been discovered fairly recently and are still topic of active research. In conclusion, I believe the manuscript requires a proper introduction that describes the state-of-the-art in wind-farm LES research, puts the presented research into perspective, and explains the flow mechanisms relevant to this work.

I thank the reviewer for pointing out the shortcomings of the introduction. I agree that the introduction contains much too less references. I now added 26 references to relevent literature and also compare the wind farm sizes of the studies. Please look at the diff document for the made changes.

In the new version of the manuscript I describe the terms "wake deflection" and "inversion layer displacement" in more detail and also added references to studies that also discover these phenomena:

"For example, large wind farms can cause a significant counterclockwise wind direction change in the wake and a vertical displacement of the inversion layer above the wind farm (Allaerts and Meyers, 2016; Lanzilao and Meyers, 2022; Maas and Raasch, 2022)."

Apart from my main comment about the introduction, I have a few more scientific and technical comments as listed below:

**Scientific comments**

1. The manuscript repeatedly talks about the size of wind farms (in terms of the rated power) considered in the simulations. The size of the wind farm is perhaps relevant for computational resources (it shows how much computational resources are available to the author), but from a physical point of view the size is irrelevant since the farms are infinitely wide in the spanwise direction. For the flow physics of infinitely wide wind farms, what matters is the streamwise length of the farm or the number of turbine rows. In this respect, I have the following specific comments:

(a) Lines 7-8, 42-43, 77-78: The rated power of infinitely wide wind farms is meaningless as it is physically irrelevant and depends on the choice of turbine spacing and spanwise extent of the numerical domain, as these two parameters determine the number of turbine columns resolved within the simulation. Please use a more relevant parameter to distinguish the different cases.

I agree with the reviewer. In the new manuscript, the wind farm length is used as parameter to distinguish the two cases. Please see the diff document for the made changes.

(b) Line 79: Why is the large wind farm simulation (16 turbine rows) twice as wide as the small one (8 turbine rows) if you are using periodic BC anyway? What is the reasoning behind this?

The idea is that at least one inertial wave amplitude fits into the domain width (Ly) for the large wind farm, so that it is well visible in Fig. 2. However, there is no physical reason for that choice. Thus, I decided to repeat the simulation for the large wind farm case using the same width as for the small wind farm (8 turbine rows) and to use the results of this simulation for the manuscript (no change in results, as expected). I adjusted Figure 1 and 2 accordingly. This also results in a reduction of computational cost, which enables me to perform the additional simulations as requested by the reviewer (longer domain and different latitude, see below).

(c) Line 92: "Note that the small wind farm is already as big as the largest wind farms of most other LES studies, e.g. Wu and Porté-Agel (2017) (length 19.6 km, rated power 0.36 GW) ..." Comparison of power for infinitely wide wind farms is meaningless because it depends on how many columns are resolved (see also previous comment). Your study has 8 and 16, Allaerts&Meyers had 9, Wu and Porté-Agel had 5...

I now only refer to the length of the wind farm.

(d) Line 48-49: "To my knowledge this is the second largest wind farm

LES study in terms of domain size and wind farm power after the study of Maas and Raasch (2022)." Why is it relevant that this is the second largest wind farm LES study? Instead of just mentioning the ranking, a quantitative comparison with wind farm size in other LES studies would be more interesting.

I deleted this sentence, because this information is already included in the sentence above (line 92).

2. The inertial wave developing in the wake of the farm is quite an interesting finding, and it leads to some strong statements related to wind speed ups and impact on downstream located wind farms (see, e.g., line 12-13 and line 173-174). If these statements hold in general, they could have strong implications for wind energy deployment. Therefore, I'm surprised to see that the study is based on only two simulations, and I share past reviewers concern about verification/validation of results. I do agree with the author that it is not so easy to turn off gravity or the Coriolis force, but there are other options to increase confidence in the presented results. For example, I assume that the amplitude of the inertial wave depends on how much the flow decelerates inside the farm, so you should be able to see different wave amplitudes with different wind farm lengths or even different wind farm layouts. Maybe it is worthwhile to consider a wind farm of intermediate length or a different layout. Alternatively, if the flow behaviour you are seeing is indeed an inertial oscillation triggered by the wind farm, the wave length should be independent of wind farm length and depend on the Coriolis parameter. You could consider a different latitude (boundary-layer height is governed by the temperature structure and subsidence so no issue there) and see whether the wave length changes accordingly.

Since there is also a variation in wind farm length I think that the variation of latitude is the best way of confirming that it is an inertial wave. Thus, I performed an additional (large wind farm) simulation with a latitude of 80°N instead of 55° N (I also used a longer domain due to the second request of the reviewer). This should result in a wavelength of 400 km instead of 480 km:

[Figure]

Because including these results into the manuscript would require heavy rewriting and because it does not affect the overall conclusion, I put this figure and a short description in the appendix B. I refer to the appendix in line 197 in the manuscript:

"To add further confidence tho this result, an additional simulation with a latitude of $80^\circ~\unit{N}$ instead of $55^\circ~\unit{N}$ is performed. The results are given in Appendix~\ref{sec.Appendix_lat} and show that the wavelength decreases to $\lambda_I=400~\unit{km}$ due to the shorter inertial period at that latitude ($T = 12~\unit{h} / \sin(80^\circ) = 12.1~\unit{h}$)."

3. I have several questions about the wavelength analysis from line 325 onward.

First of all, I believe equation 14 is incorrect. Based on equation 12 and 13 and assuming that you define the "absolute wavelength λ" as the wave length in the direction of phase propagation, i.e. $\lambda = \lambda x \cos \alpha$ (note that this is nowhere defined!), I find that the absolute wavelength should be given by

$$\lambda = \frac{1}{\sqrt{1 + \frac{f^2 \sin^2 \alpha}{N^2 \cos^2 \alpha}}} \frac{2\pi U}{N}$$

Equation 14 has now been corrected. The definition of the absolute wavelength is now included in the text and in equation 14:

lines). The absolute wavelength $\lambda$, i.e. the wavelength in the direction of phase propagation, is then given by

$$\quad \lambda = \frac{2\pi c}{\omega} = \frac{2\pi U \cos\alpha}{\sqrt{f^2 \sin^2\alpha + N^2\cos^2\alpha}} = \frac{1}{\sqrt{1 + \frac{f^2\sin^2\alpha}{N^2\cos^2\alpha}}}\frac{2\pi U}{N}, \tag{14}$$

I checked the equations in the spreadsheet that generates Table 1 and they are correct.

Second, looking at the first two entries in table 1, how is it possible that λ for wave type 1 and 2 (small wind farm) is the same for two different inclination angles α? According to eq. 14, there should be a unique relation between α and λ for U, N and f constant.

That is true. It is just the result of rounding. The wavelength is approximately constant up to an angle of 85° and changes rapidly in the last 5°:

[Figure]

The numbers have now more digits so that the differences are visible:

| Wave type | $\alpha$ | $\lambda$ | $\lambda_x$ | $\lambda_z$ | $\lambda_x/L_{wf}$ | $L_{wf}/L_s$ |
|---|---|---|---|---|---|---|
| | ° | km | km | km | – | – |
| 1 (LE + TE) | 60 | 5.29 | 10.6 | 6.11 | – | – |
| 2 (small wind farm) | 83.7 | 5.27 | 48.0 | 5.30 | 3.6 | 2.5 |
| 2 (large wind farm) | 88.3 | 4.96 | 167.1 | 4.96 | 1.85 | 17.0 |
| 3 (wake) | 89.3 | 3.91 | 320.3 | 3.91 | – | – |

**Table 1.** Inclination angle of phase lines $\alpha$ and corresponding wavelengths for the different wave types present in Fig.9.

Thirdly, eq. 14 has two unknowns: the inclination angle and the wave length. The manuscript does not mention how you come to the results in table 1? Did you measure the wavelength in the simulation and then calculated the inclination angle? Or did you approach it the other way around, estimating the inclination angle from the figures and then calculating the wave length based on eq. 14?

This information is now included in the text:

"The inclination angles of each wave type are measured in a Figure that is similar to Fig. 9 but uses equal scales for both axes (not shown). The calculated oscillation frequencies and wavelengths of the three wave types are listed in Table 1."

**Minor/technical comments**

1. Eq.3: In the last term on the right-hand side, the subscript of u should
be j instead of i.

Is corrected.

2. Line 105: "... which is enough for resolving the gravity waves with a wave
length of approximately 5 km." How did you calculate that wave length?
This is coming out of nowhere. Please explain, or refer to later section.

I added a reference to Table 1 and use **vertical** wavelength instead of just wavelength:

"Above 900 m the grid is vertically stretched by 8 % per grid point up to a maximum vertical grid spacing of 200 m, which is enough for resolving the gravity waves with a **vertical** wavelength of approximately 5 km **(see Table 1 in Sec. 3.4)**."

Additionally, I corrected "wave length" to "wavelength", also at two other locations in the text.

3. Can you briefly describe the radiation boundary conditions of Miller and
Thorpe (1981) and Orlanski (1976) in the paper?

I have added two sentences and an equation describing the radiation boundary condition:

velocity fields, for further details please refer to Maas and Raasch (2022) and Munters et al. (2016). Radiation boundary conditions as described by Miller and Thorpe (1981) and Orlanski (1976) are used at the outflow plane. Hereby, the flow

125 quantity $q$ at the outflow boundary $b$ is determined with the phase velocity $\hat{c}$ and the upstream-derivative of the flow quantity:

$$q_b^{t+\Delta t} = q_b^t - (\hat{c}\Delta t/\Delta x)(q_b^n - q_{b-1}^n). \tag{5}$$

The phase velocity $\hat{c}$ is set to the maximum possible phase velocity of $\Delta x/\Delta t$. The surface boundary conditions and other parameters are the same as in the precursor simulation and are thus described in the next section. The physical simulation time

As the inertial oscillation
triggered by the wind farms is a wave phenomenon itself, with a very large
wave length (see figure 2 and 3), how do you know that the flow results
are not affected by the outflow boundary? Clearly, in figure 3a, the wave
extends all the way down to the outflow boundary. If you would put
that boundary closer or further away, do you still obtain the same wave
properties?

I performed an additional large wind farm simulation with a domain length of 655.36 km instead of 409.60 km. The resulting wind speed, wind direction and pressure evolution along x is shown below. As can be seen, there are only very small changes that do not lead to different conclusions. Especially, the flow directly at the (old) outflow boundary (x=409.6 km) is not affected. The results are part of the new appendix B that also shows the variation of latitude.

[Figure]

4. Line 119: What values are used to come to the advection distance of the convective time scale? Also, please explicitly mention the definition of the convective velocity scale w and the values used to calculate the quantity.

This information is now included in the text:

plane at $x = 0$. Details of the recycling method are given in Maas and Raasch (2022). The large distance between inflow and recycling plane is chosen to cover elongated convection rolls that appear in the CBL and to cover at least twice the advection distance of the convective time scale $U_g z_i / w_* = 9.011$ ms$^{-1}$ 600 m/0.49 m s$^{-1}$ $\approx$ 11 km, with the convective

120    velocity scale $w_* = \left[ \frac{g z_i}{\theta} \langle \overline{w'\theta'} \rangle_s \right]^{1/3}$, where $\bar{\theta} = 280$ K and $\langle \overline{w'\theta'} \rangle_s$ is the horizontally averaged kinematic surface heat flux averaged over the last 4 h of the precursor simulation. For the potential temperature, the absolute value is recycled instead of

5. Section 2.3: Please explain the particular choice of surface heating and boundary-layer height (for instance, a boundary-layer height of 600m in convective conditions seems quite low).

The choice of parameters, also the lapse rate, is now justified shortly:

speed of $9.0~\text{ms}^{-1}$ that is aligned with the x-axis. The latitude is $\psi = 55°$ N. The initial potential temperature is set to 280 K up

to a height of 600 m and has a lapse rate of $\Gamma = +3.5~\text{K}\,\text{km}^{-1}$ above. This lapse rate corresponds to the international standard atmosphere. The onset of turbulence is triggered by small random perturbations in the horizontal velocity field below a height of 300 m. A Dirichlet boundary condition is set for the surface temperature. Why a Dirichlet boundary condition is a good choice is explained in Maas and Raasch (2022). A constant surface heating rate of $\dot{\theta}_0 = +0.05~\text{K}\,\text{h}^{-1}$ is applied, resulting in a Monin-Obukhov length of $L \approx -400$ m, which is common value for convective boundary layers in the North Sea (Muñoz-Esparza

et al., 2012). The resulting boundary layer height (height of maximum vertical potential temperature gradient) of $z_i = 600$ m is a small but still typical value for convective boundary layers over the North Sea (Maas and Raasch, 2022). Boundary

I agree with the reviewer that a boundary layer height of 600 m seems to be small for CBLs over the sea. However, to my knowledge, there are no published long term measurements of the boundary layer height over the North sea. Due to this lack of measurements, I used the boundary layer height output of the COSMO model in my last paper (Maas and Raasch 2022) to generate a climatology (see the appendix of that paper). Based on this climatology, 600 m is still a very common value. I added a remark in the conlcusions, that a change in BL height may affect the results significantly (line 567):

"Additionally, only one meteorological setup is used in this study. A change in BL height, stability or wind speed may affect the results significantly."

6. Line 130-132: "The initial horizontal velocity is set to the geostrophic wind (Ug,Vg) = (9.011,-1.527) ms-1, resulting in a steady-state hub height wind speed of 9.0 ms-1 that is aligned with the x-axis." How did you find this particular geostrophic wind speed and direction? Did you find this by trial and error or did you use a particular method (e.g., a wind speed/angle controller)?

I have added this information:

"The initial horizontal velocity is set to the geostrophic wind $(U_g , V_g )$ = (9.011, −1.527) ms−1, resulting in a steady-state hub height wind speed of 9.0 ± 0.02 ms−1 that is aligned with the x-axis (±0.01 ∘ ). The values for the geostrophic wind are obtained by iterative adjustments between preliminary precursor simulations, of which two are needed to obtain the given accuracy."

7. Line 144: "... so that the inertial oscillation has decayed ..." I assume you are referring to an inertial oscillation in time that occurs after the simulation is initialized? Please explain why an inertial oscillation is triggered. Please also clearly mention that you are talking about an inertial oscillation in time to avoid confusion with the inertial oscillation observed downstream of large wind farms.

"The physical simulation time of the precursor simulation is $48~\unit{h}$, to obtain a steady state mean flow, i.e. the hourly-averaged hub height wind speed changes less than $0.05~\unit{m s^{-1}}$ within $8~\unit{h}$. This long simulation time is needed for the decay of an inertial oscillation in time that has a period of $14.6~\unit{h}$. The inertial oscillation occurs, because there is no equilibrium of forces in the BL at the beginning of the simulation."

8. Line 182: "Further downstream the wind turn clockwise ..." → "Further downstream, the wind turns clockwise ..."

Corrected.

9. Line 209: "Because the wind farms are infinite in the y-direction, the perturbation pressure gradient force is parallel to the x-axis and has thus no effect on the wind direction at first." Is this because the perturbation pressure is due to gravity waves, which are uniform in y direction because the wind farm is infinitely wide? This statement needs more explanation.

Yes, I added this information in the sentence:
"Because the wind farms are infinite in the $y$-direction, the gravity waves are uniform in the $y$-direction and thus the perturbation pressure gradient force is parallel to the $x$-axis and has no effect on the wind direction at first."

10. Line 220: "If the Rossby number [based on the wind farm length] is smaller or close to 1, Coriolis effects become dominant ...". It is not necessarily true that Coriorlis forces become dominant at Rossby number close to 1. I agree that Coriolis forces become dominant as the Rossby number decreases, but you cannot claim (at least not based on two simulations) that the tipping point is for Rossby equal to 1.

That is true. It is rather the order of magnitude of the Rossby number that matters. I have rephrased this part accordingly:
"An inertial wave occurs if the Rossby number has the order of magnitude of $1$ or smaller. Coriolis effects become more dominant for smaller Rossby numbers so that the amplitude of the inertial wave is larger for the large wind farm ($Ro = 0.8$) than for the small wind farm ($Ro=5.0$)."

11. Figure 6: It is surprising to see that neither TKE nor TI shows an oscillation. I would expect at least one of the two to be affected by the oscillatory behaviour in wind speed, as TI is related to TKE normalized by that same wind speed. How do you explain this?

I extended the x-axis of the figure so that the entire domain length is captured. As can be seen, there is a very small oscillation in TKE. However, this is hardly visible in Fig. 6 because the turbine-induced TKE level is much higher than the TKE level in the far wake. I added a sentence on that in the manuscript:

"Further downstream, the TKE and the TI also show a slight oscillation as the wind speed and direction show (see Fig. 6a and b). However, the amplitude is much smaller than the TKE and TI levels that occur inside the wind farms and thus the oscillations are hardly visible"

[Figure]

Further, on lines 235-236 you say that "TKE is greater in the small wind farm, because the wind speed is greater." How is TKE related to the wind speed magnitude? TKE production is related to wind speed gradients, so I don't see how TKE is directly related to wind speed magnitude.

Yes, in this case it is turbine generated TKE, which is wind speed dependent. I expanded the sentence:
"because the wind speed is greater and thus the turbines generate more TKE."

12. In lines 248-250 you say that "TI does not show the oscillatory behavior that the wind speed and direction show because turbulence has time scales that are orders of magnitude smaller than that of the mean flow and therefore hardly affected by the Coriolis force." Are you saying that the turbulence time scales are orders of magnitude smaller than the inertial oscillation period, and that therefore TKE rapidly adapts to changes in wind speed such that TI (or TKE) is constant? Please clarify this statement.

I deleted this sentence, because TKE and TI show slight variations due to the change in wind speed (see 2 comments further above).

13. Line 252: "The last two sections ..." → "The previous two sections ..."

Corrected.

14. Line 349: "...which approximately corresponds the the length of the small

wind farm ...” should be ”...which approximately corresponds to the length of the small wind farm ...”

Corrected.

15. Line 356: ”Due to the large ratio of Lwf /Ls in the large wind farm case, wave type one and two can be clearly distinguished in this study.” Why would a larger ratio of wind farm length to Scorer parameter make the wave more distinguishable?

Due to the larger seperation distance in terms of wavelengths. Rephrased:
"Due to the large ratio of $L_{wf} / L_s$ in the large wind farm case, the waves at the wind farm LE and TE (type one) are separated by several wavelengths and can thus be clearly distinguished from wave type two in this study."

16. Section 3.5.1: How did you scale the budget terms from $W\rho^{-1}$ to MW per turbine? Did you assume $\rho$ = 1.17 kg m−3 like in eq. 11?

Yes, I have added this information:
"The air density is $\rho = 1.17~\unit{kg~m^{-3}}$."

17. Caption of figure A1: apprxomation $\rightarrow$ approximation

Corrected.